# The evolution of aquaculture in the Mediterranean region: An anthropogenic climax stage?

Benedetto Sicuro *

Department of Veterinary sciences, University of Torino, Grugliasco (TO), Italy

* benedetto.sicuro@unito.it

## Abstract

This study is the investigation of Mediterranean aquaculture complete history, from 1950 to 2020. Both functional than geographical expansion of aquaculture is investigated, considering two main complementary aspects of aquaculture: farmed species and farming countries. According to the models proposed in this research, Nile tilapia and Egypt will dominate the future of Mediterranean aquaculture. Malta and Israel are the first producer countries, in relative terms. The most pervasive species are European sea bass and gilthead sea bream that are promising for a future expansion. In several countries, aquaculture has huge potentiality of development and it could grow with a factor of 5 or more, based on the ratio capture vs fishery on country size. Aquaculture total production in 2020 was of 2.8 Mln tons and it is expected to reach from 3.65 Mln tons in 2030. Aquaculture will grow in the countries and species that in this moment are dominant and the future of Mediterranean aquaculture will be characterized by the affirmation of these ones.

**Data Availability Statement:** All relevant data are within the manuscript and its Supporting Information files

## Introduction

Mediterranean region is a hot spot of marine diversity and it has a specific biogeographical value, in fact Mediterranean ecoregion is a biome with similar conditions in California, South Africa and South America [1]. Marine resources represent a traditional source of sustainment in this region, which borders are extended along 45,830 km of coastline and all the countries surrounding Mediterranean Sea share its natural resources [2, 3]. Until few years ago the exploitation of these aquatic resources has been essentially based on capture fishery [4] and the rooted tradition of seafood consumption in this region [5, 6] has caused a tremendous development of capture fishery that currently represents a serious threat to the maintenance of the wild fish stocks [7, 8]. Currently, fisheries and aquaculture provide food and employment to coastal countries employing about 600,000 people [4, 9–11], in particular in the Southern coast of Mediterranean [12–15]. Aquaculture in this region represents a social and economic opportunity [16–21] and its future will follow the Sustainable Development Goals (SDGs) indicated by United Nations [10]. Moreover, Europe is considered one of the largest seafood markets in the world and this is an optimal opportunity of development for the southern Mediterranean

**Funding:** The author has no received specific funding for this work.

**Competing interests:** The authors have declared that no competing interests exist.

countries [22]. Aquaculture productions are growing all over the world, but in each continent and region there are specific differences that are the effect of biological, geographical, and socio/political features. Of course, past productions cannot be considered as a unique method to estimate aquaculture future, as socio/political and cultural factors certainly affect aquaculture productions [23], but the 67 years of production data analyzed here represent the entire history of Mediterranean aquaculture. Aquaculture is an economic relevant productive activity in the region with positive perspectives [2, 10, 24–26] and a critical analysis of the history of aquaculture in the region enlightens key factors for aquaculture progress and suggests an original interpretation of analyzed situation.

Aquaculture is originated by capture fishery, and diffusion of farmed species can be interpreted as the expansion of a group species in an unexploited territory. An evolutionary approach inserts aquaculture in a general theoretical context that opens original interpretations for the future of aquaculture, thus showing similitudes with natural phenomena. Aquaculture in ecological terms, is a kind of modern colonization of aquatic ecosystems [27].

The aim of this paper is: a) to investigate temporal and geographical expansion of aquaculture in the Mediterranean region; b) to give valuable perspectives for its future development, thus inserting aquaculture in an evolutionary context.

## Materials and methods

Mediterranean data about aquaculture productions have been collected since 1950 [24, 25, 28, 29]. Data utilized have been extracted by FAO aquaculture databases (www.fao.org/fishery/statistics/software/fishstatj/en, updated 2020) and are the largest existing database in aquaculture in this region [30–33]. Both multivariate and regression methods have been used for data elaboration. For the statistical elaboration of temporal data, regression has been utilized, with polynomial and/or linear models indicated in the figures, together with Pearson correlation coefficients and regression coefficients. For cluster analysis, "Cluster" package of R software [34, 35], was used, dendrograms have been calculated with Euclidean distances and Ward Hierarchical Clustering method. For heatmaps, "Pheatmap" package of R software, was used, dendrograms have been calculated with Euclidean distances and cluster centroid for clustering method. Dendrograms have been calculated only on the rows that are fish species, as the focus of the data analysis was the study of temporal succession of farmed species in the Mediterranean. Multiple correlation plot has been obtained with R package "Performance Analytics". In the multiple correlation plots, frequency histograms have been added. Thematic maps have been obtained with R package "rworldmap".

Shannon index (H) has been calculated as follows:

$$H = -\sum_{n=1}^{s}(p_i * ln(p_i)$$

where $p_i$ is the percentage of production volume of each species in each year (i); S is the total number of the species.

The Pareto Index [27] has been calculated as follows: annual ratio between dominant and relevant species number, where relevant species are those accounting for 99% of annual production and dominant ones are those responsible for 80% of production, based on relevant species. Minor species are those accounting for less than 1% of annual production, obtained by the difference between total and relevant species. (As example: in 1950 10 species were totally farmed, of which 7 accounted for 99% of annual production, relevant species. On these 7, 2 accounted for 80% of production, dominant species. Minor species were 3, the difference between total and relevant species. Pareto index was ratio between dominant and relevant, *i.e.*

28.6%). Minor species have been excluded by the calculation of this index, as their contribution is irrelevant to aquaculture diversity and can be eliminated. Minor species proliferation in aquaculture is the effect of continuous trials and errors mechanism typical of any modern technological activity, thus the approximation proposed in this index put the focus on the core of diversity. All statistical analyses were conducted in R R version 4.1.0 (2021-05-18) — "Camp Pontanezen" Copyright (C) 2021 The R Foundation for Statistical Computing Platform.

## Results

### Aquaculture volumes and main species

Mediterranean aquaculture production in 2020 was about 2.8 Mln tons of volume for a corresponding value of about 8 billion US$. The calculated models (Figs 1 and 2) clearly indicate ($R^2 > 0.98$) a noticeable growth, with a prevision of about 3,65 Mln tons in 2030 and an average annual growth rate of 5%, both in the last 5 than in the last 10 years.

The model based on the entire considered period indicates a quasi-exponential growth and considering the period of time of last 10 years (Fig 1), it is clear that capture fishery volume is constant about a 4 Mln tons/year, instead aquaculture is still growing. In fact, the linear model calculated (Fig 1) indicates an estimated volume of about 4.2 Mln tons for 2030. The ratio between aquaculture and fishery volumes shows that in the 1950 capture fishery volume was 130 times that of aquaculture while in the last years this ratio is reduced to 2 and expected to be close to 1 in the next 10 years (Fig 2).

The direct comparison of capture fishery and aquaculture in the region confirms that fishery volume has reached a maximum value at the end of 80ies and after that period only aquaculture production has increased (Figs 1–4). The average value of capture fishery volumes between 1985 and 2020 (4,15 Mln tons) can be considered the Mediterranean carrying capacity. According to these models, aquaculture production should reach in 2030 the same amount of fishery carrying capacity, *i.e.* 4,2 Mln tons.

On the base of cumulate volumes of productions, farmed species can be divided in 3 groups. Nile tilapia is the main species in the region (Fig 5), a second group of species that contain both the marine species than the traditional species of Northern countries, such as rainbow

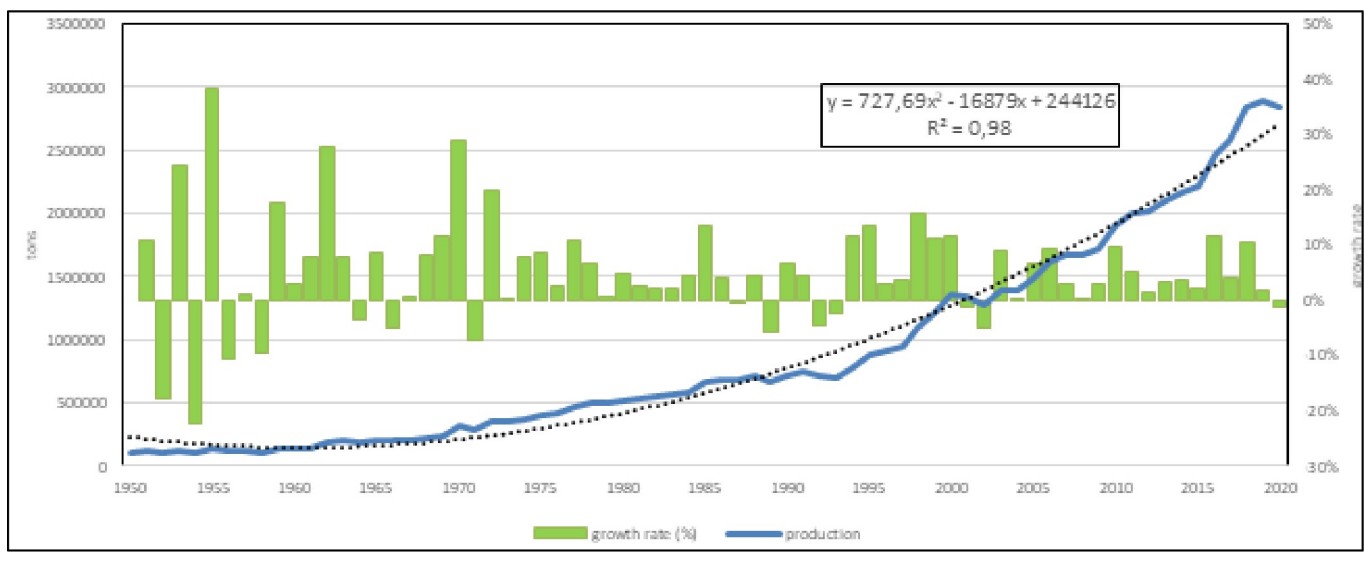

**Fig 1. Mediterranean aquaculture productions and annual growth rate from 1950 to 2020.**

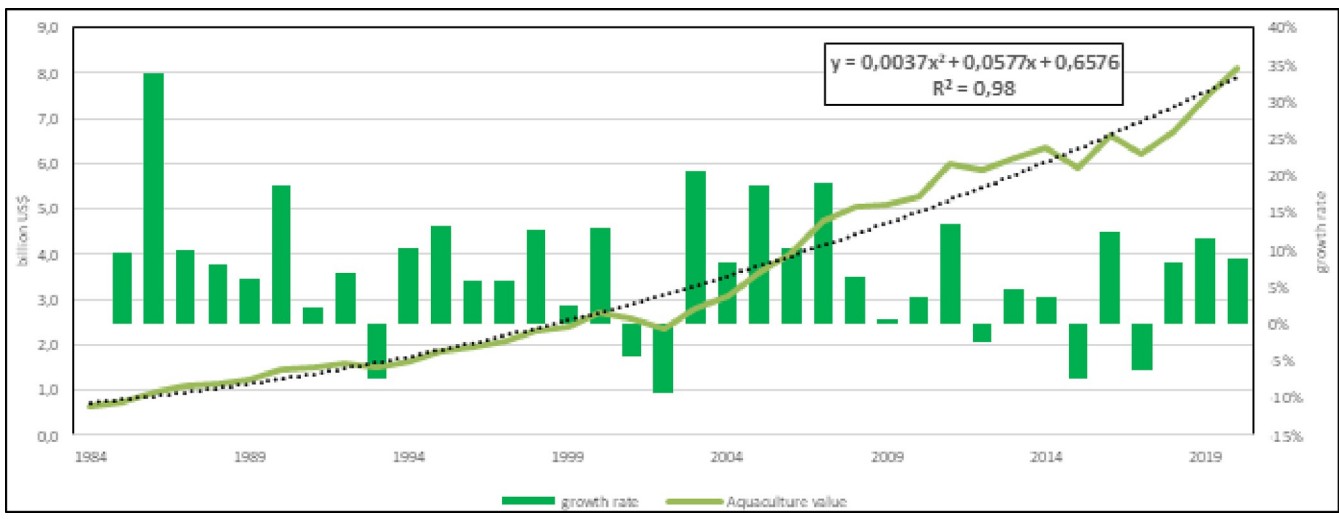

**Fig 2. Mediterranean aquaculture economic value and annual growth rate from 1984 to 2020.** Dotted line: polynomial regression line, with equation and regression coefficients.

trout. First two clusters represent main species, while the third group of species contain most of species of extensive aquaculture. The importance of rainbow trout has been largely underestimated in this region that is currently characterized by marine aquaculture, instead, rainbow trout has been one of pioneer aquaculture species in the region. Rainbow trout is a fundamental species in the Northern countries, and it is one of the first farmed species of European aquaculture, together with blue mussels (Figs 5 and 6) [6].

**Temporal and spatial diffusion of farmed species.** Although volumes represent the size of aquaculture, two other factors should be investigated: temporal and spatial expansion of aquaculture (Fig 6). Temporal expansion of aquaculture is the number of years of presence of a certain species, its persistence; spatial diffusion of aquaculture is the number of countries where a certain species is farmed, its diffusiveness.

Firstly, there is a group of species farmed from more than 60 years (Fig 6): Nile tilapia, blue mussels, rainbow trout, silver carp, eel and flathead grey mullet that make the historical core of Mediterranean aquaculture. These species occupy a stable aquaculture niche in Mediterranean region, even if they are farmed in different volumes, they represent a stable community of species, comparable with a species community in a natural ecosystem. Gilthead seabream and European seabass represent the effect of innovation in aquaculture and the pervasiveness of technology, since its introduction in Mediterranean aquaculture they rapidly spread in several countries, they are the most diffuse species in the region [3, 36–39] as they are currently farmed in more than 20 countries with relevant volumes. Nile tilapia has a limited geographical diffusion, that is mainly farmed in Egypt. The species farmed in less than 15 countries and from a period of time comprised within 15 and 40 years can be considered the potentially emergent species for aquaculture, as meagre, mullets and oysters.

The geographical expansion of farmed species must also include the expansion rate that is a measure of geographical potentiality of the species. The Geographical Expansion Rate (GER) here proposed, represent a quick indication of potential geographical expansion. In the Mediterranean region (Fig 7), few species have been farmed in more than 15 countries: European sea bass, gilthead seabream, eel, rainbow trout and common carp. Nine species out of these sixteen, show positive GER values (Fig 7), thus indicating a potential geographical expansion. Other species are of local importance and some of these do not show perspectives of

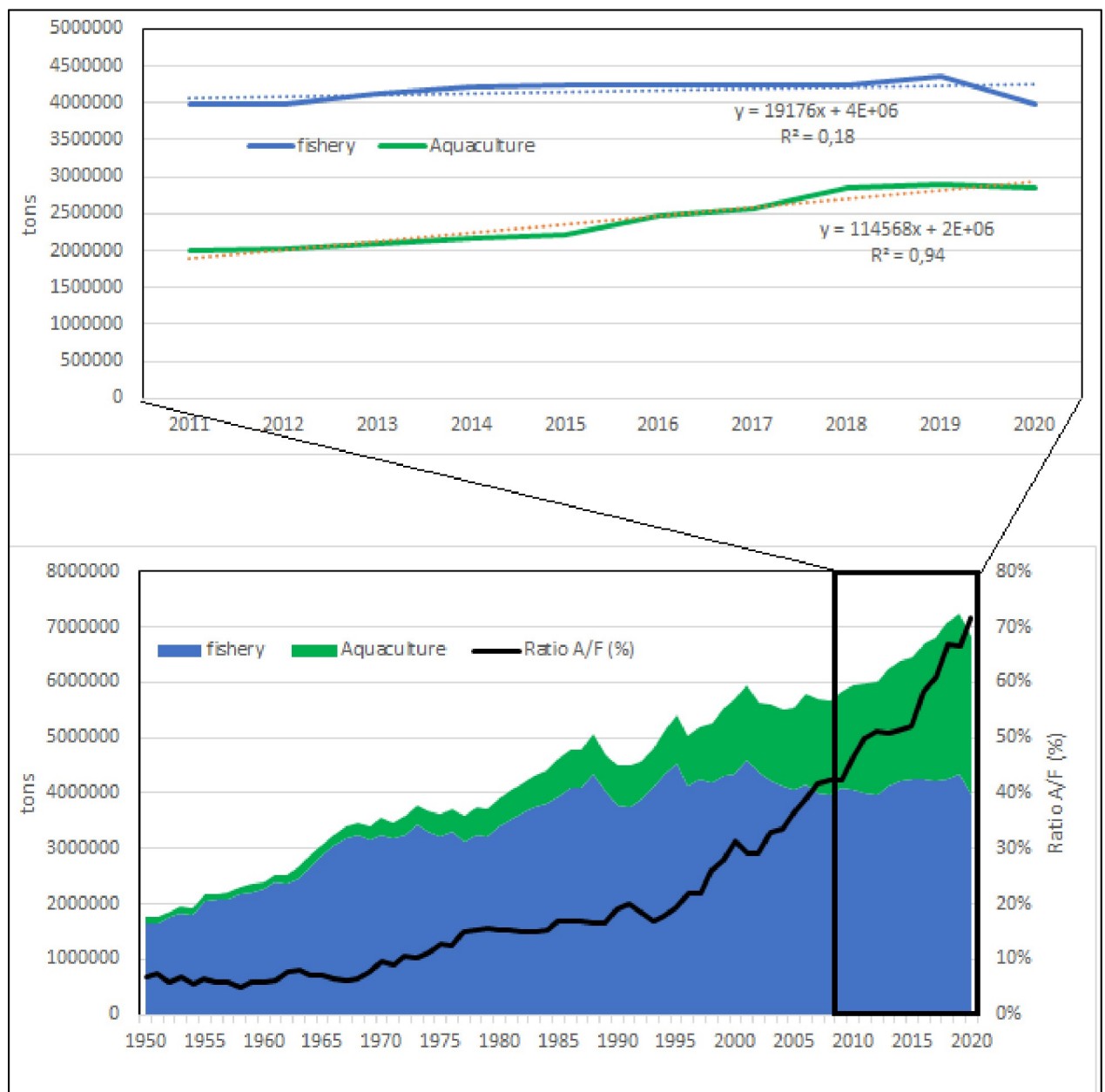

**Fig 3. Fishery and aquaculture productions in the Mediterranean.** Black line, bottom: ratio aquaculture/fishery volumes. Dotted line, top: polynomial and linear regression lines, with relative equations and regression coefficients.

geographical expansion, as grass carp, silver carp and blue mussels. Meagre (*Argyrosomus regius*) is the species with better perspectives of geographical expansion [40, 41]. Geographical expansion of sea bass, seabream and meagre show peaks related to innovation diffusion [36, 37, 41, 42] while species of extensive aquaculture [12] have a gradual expansion caused by translocation of more simple farming techniques as the case of bivalves.

Marine aquaculture which includes brackish water production of lagoons spread all along the of Mediterranean coasts, is the main sector of aquaculture (Figs 8 and 9), as expected [5], reaching almost 2 Mln tons of production in 2017, and it is rapidly growing. Total production

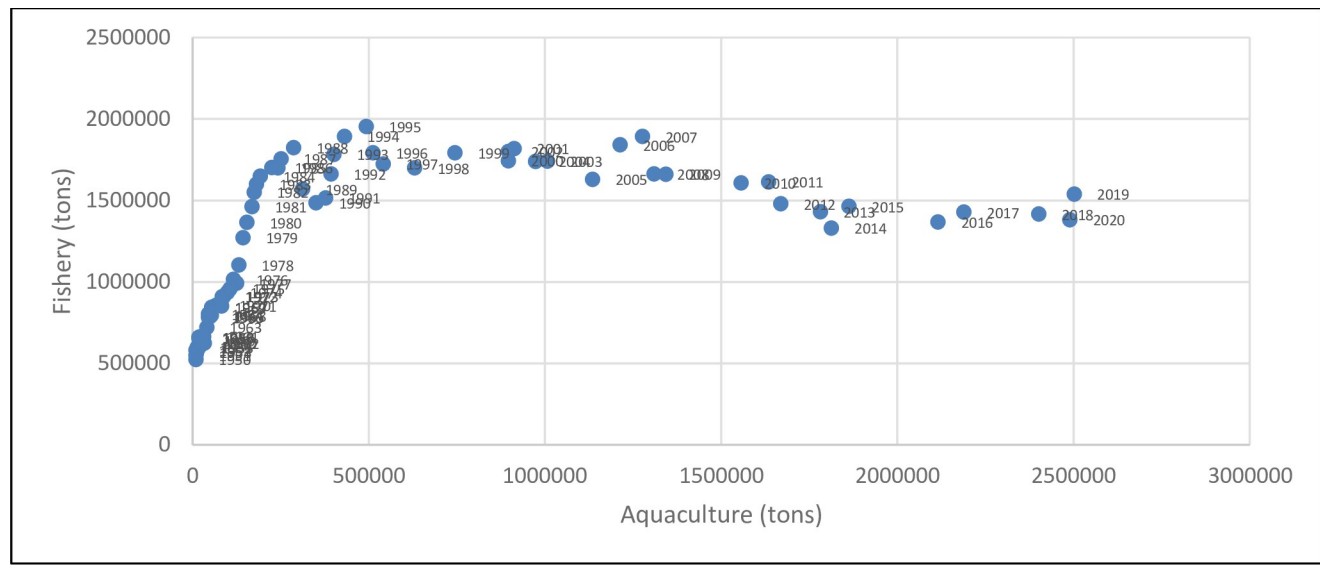

**Fig 4. Capture fishery and aquaculture annual productions (considering only Mediterranean coastline of respective countries).**

of freshwater aquaculture reached 0,5 Mln in 2017 tons and it showed some negative variations after 2005 that could indicate a stabilization of production. On the base of calculated models, it should reach about 0.9 Mln tons in 2030. The higher production of marine aquaculture product is perfectly explainable by the traditional sea food consumption in the region [43–45].

Fish and bivalves [46] are the main zoological groups of Mediterranean aquaculture (Figs 10 and 11) with negligible productions of crustaceans and algae. There are few algal species candidates for aquaculture, Gracilaria has been farmed in the past and Spirulina is the main species. The total production of Spirulina was of 302 tons in 2017 of which 150 tons in Greece.

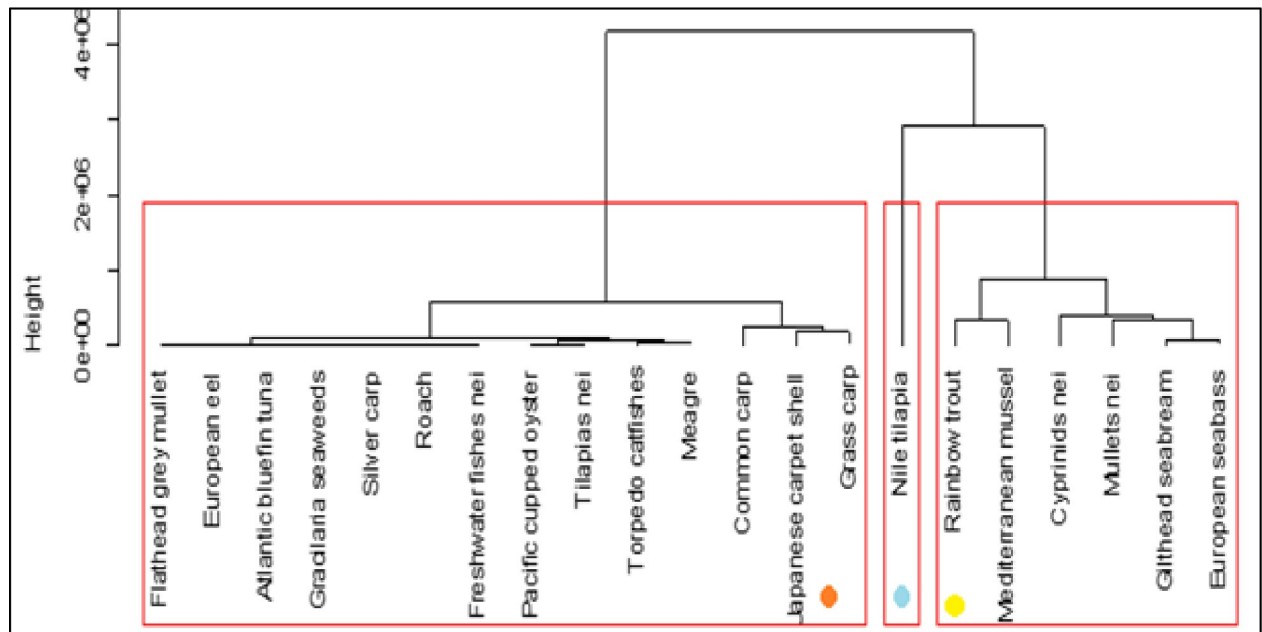

**Fig 5. Cluster of main species of Mediterranean aquaculture, based on annual productions (calculated with Euclidean distances).**

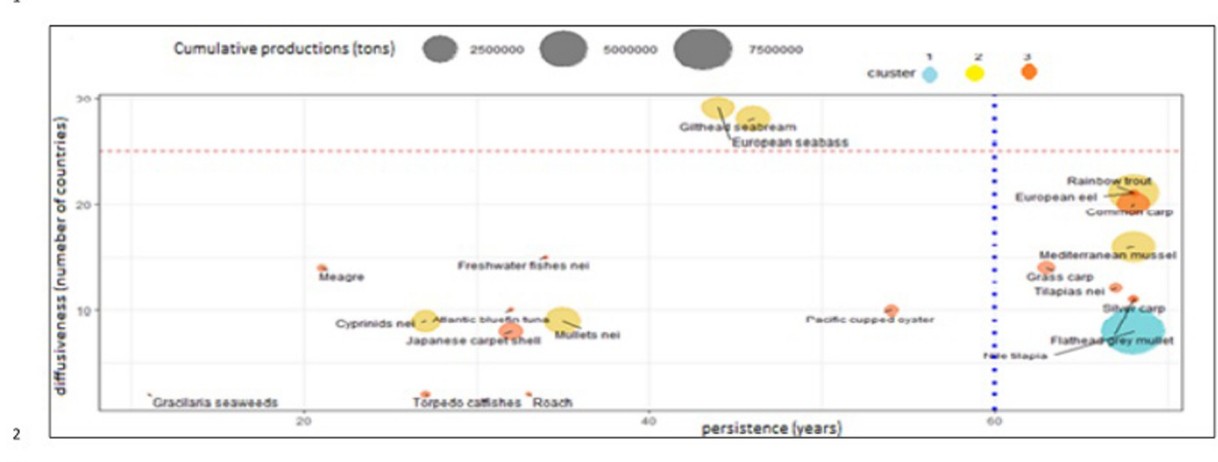

**Fig 6. The main species farmed in the Mediterranean region, considering cumulate productions, number of countries where the species is farmed (diffusiveness) and number of years of farming (persistence).** Vertical line separates the species farmed from more than 60 years. Horizontal line separates species farmed in more than 20 countries.

Bivalve farming is a form of traditional extensive aquaculture [47] based on blue mussel productions in Italy and in Greece where it is declining in the last years [46] even if there is a rooted tradition of consumption [48–50]. Small volumes of production make more evident variation in the growth rate, due to the starting of this activity in new countries or to the difficulties related to seed production [4]. The total amount of crustacean farmed in 2020 was 5270 tons, of which 2100 tons whiteleg shrimp produced in Egypt [51, 52].

Fish farming is the main production in the Mediterranean [6, 53, 54], in fact in 2020 accounted for more than the 90% of entire aquaculture production in the region. Within fish species, Mediterranean aquaculture has 4 species with potential further growth: Nile tilapia, European sea bass, gilthead seabream and in less extent, mullets (Figs 12–15) [9, 38, 42, 43, 45, 55]. According to the calculated models, Nile tilapia is expected to reach a production about 1.3 Mln tons in 2030, but observing the reduction of growth rate in the last years, it can be supposed a stable production about 1 Mln tons that is the average production of the last ten years. The production models calculated on the complete time series indicate that European sea bass and gilthead sea bream are expected to reach 339000 tons (for a value of 1.85 US$ billions) and 326000 tons (for a value of 1.76 US$ billions) in 2030, respectively. However, the growth rate reduction in the last 10 years indicates a stable production for these species about 100000–120000 tons per species (for a corresponding value of 590–610 US$ Mln). Mullets production trend has showed some oscillations from 2008 to 2010 due to heavy variations registered in Egypt, and the estimated annual productions for next years, based on growth rate on the last 10 years, are of 100000 to 120000 tons, thus comparable with sea bass and sea bream. Even in the case of mullets, the zoological division in species does not correspond to a functional separation of species from the point of view of aquaculture. There are 21 species of mullets worldwide spread (mugilidae family) of which 5 or 7 of interest for aquaculture in the Mediterranean, farmed in similar conditions and hardly distinguishable between them, they should be considered as a super - species.

**Species succession.** During the 70 years examined, aquaculture productions are not only changed in terms of volumes but also in terms of relative composition [54].

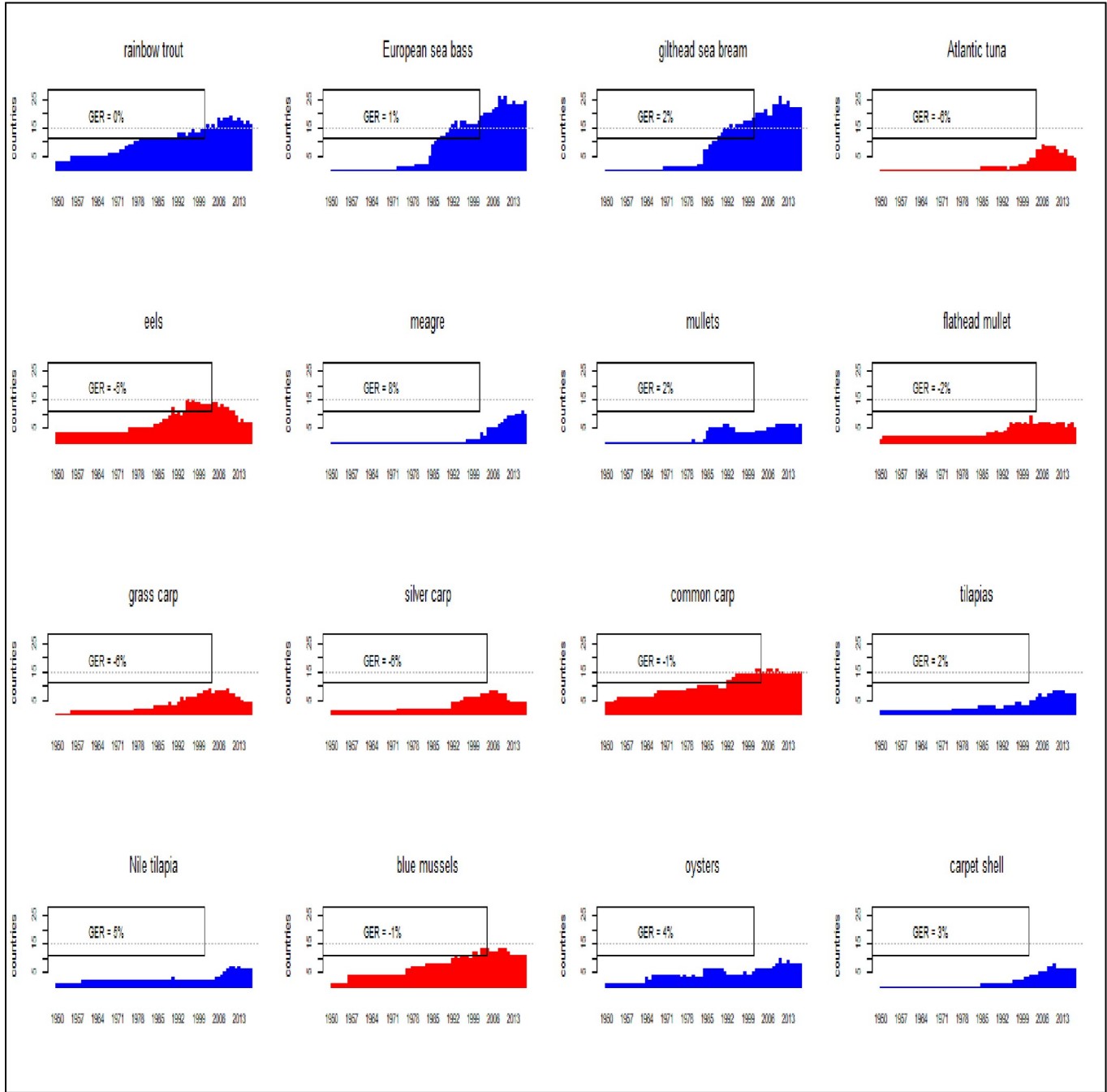

**Fig 7. Geographical expansion of main species farmed in the Mediterranean region.** Horizontal line indicates 15 countries. Geographical Expansion Rate (GER): annual expansion rate (%), calculated as average of the last 10 years 2017–2008. Blue histograms (positive GER) indicate species in geographical expansion; red histograms (negative GER) indicate species in geographical contraction.

An analysis of species evolution exclusively based on production volumes (Fig 16) shows a progressive affirmation of Nile tilapia that in 2020 was the 53% of entire Mediterranean aquaculture production. The huge production of Nile tilapia in Egypt conceals the real nature of Mediterranean aquaculture (Fig 16), as happens for China in global aquaculture productions. For this reason, a relative representation of species succession can enlighten the functional

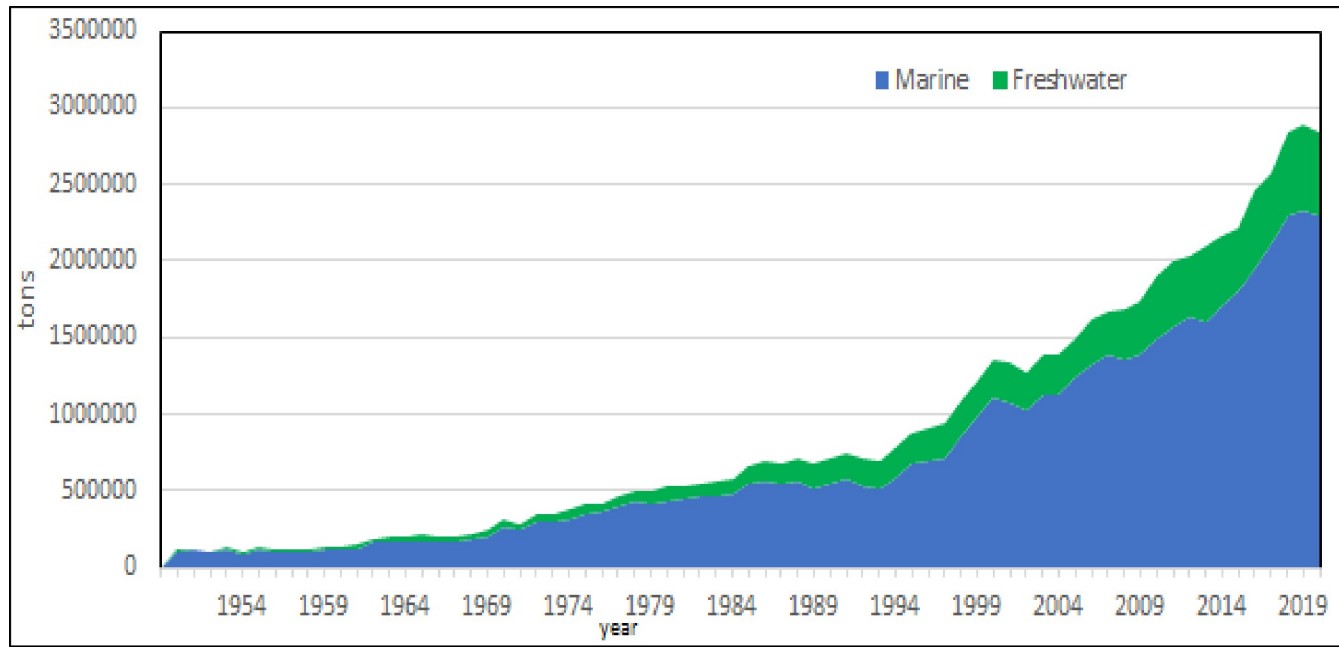

**Fig 8. Evolution of two main aquaculture sectors: Marine and freshwater aquaculture.**

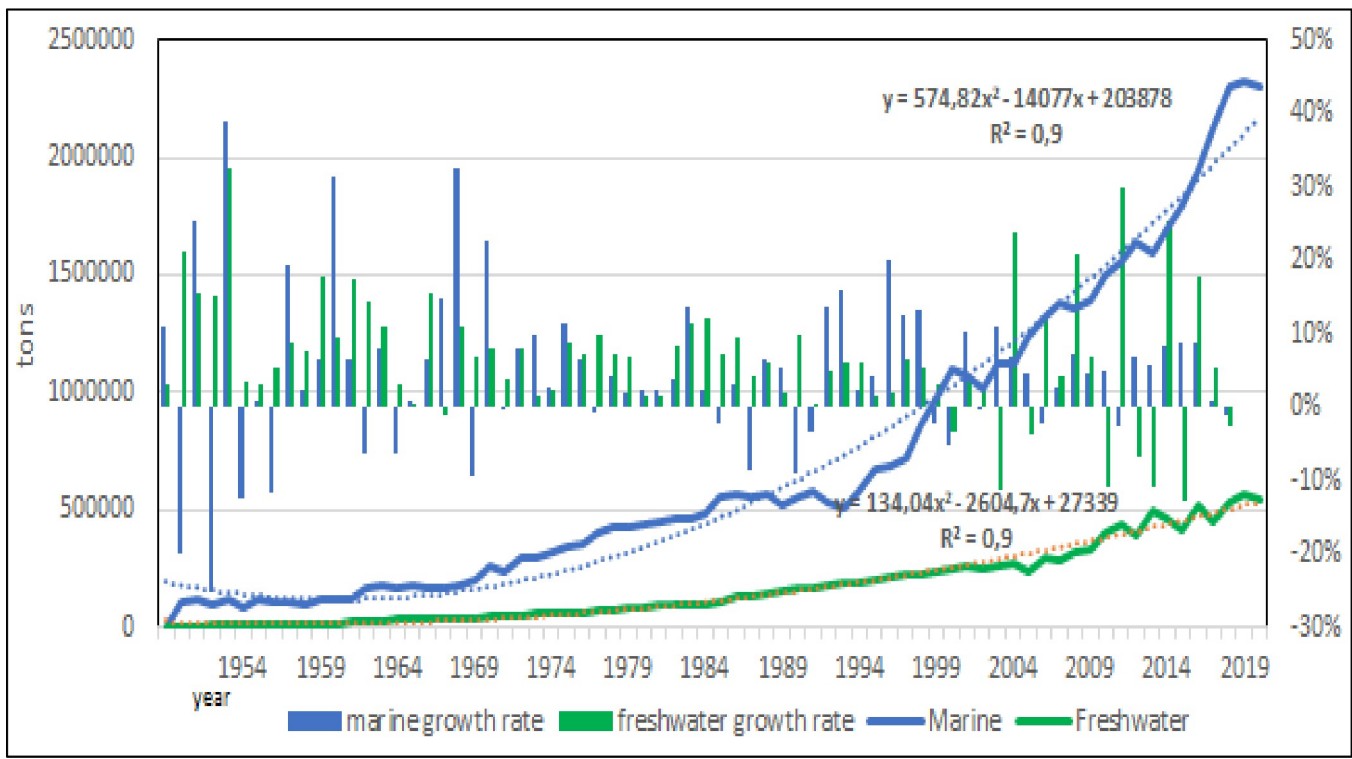

**Fig 9. Evolution of two main aquaculture sectors: Marine and freshwater aquaculture: Annual growth rate.** Dotted lines: polynomial regression lines, with relative equations and regression coefficients.

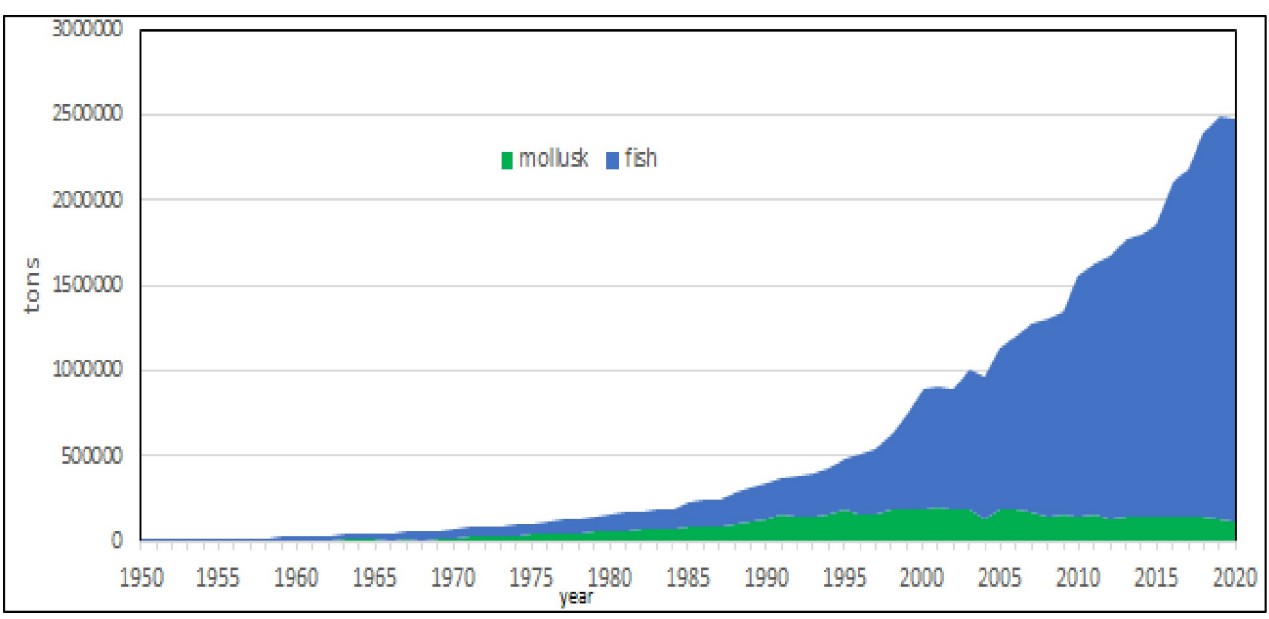

**Fig 10. Fish and bivalve farming in the Mediterranean region.**

succession of species in the Mediterranean (Fig 17). Accordingly, three main successive phases can be observed (Fig 17). From 1950 to 1970 common carp was dominant species. Common carp was the most diffuse species in aquaculture at that time and a pioneer species of fish farming worldwide. Successively, roughly from 1970 to 2000, dominant species were blue mussels

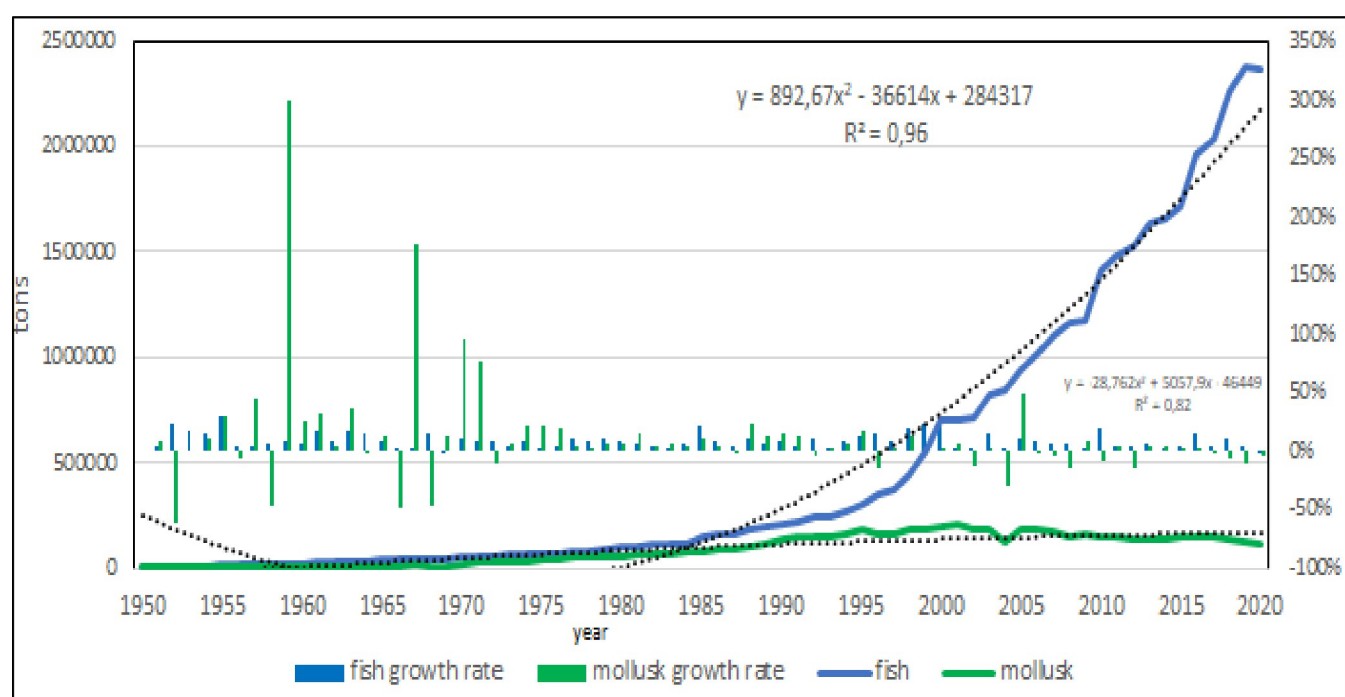

**Fig 11. Fish and bivalve farming in the Mediterranean region: Growth rate.** Dotted lines: polynomial regression lines, with relative equations and regression coefficients.

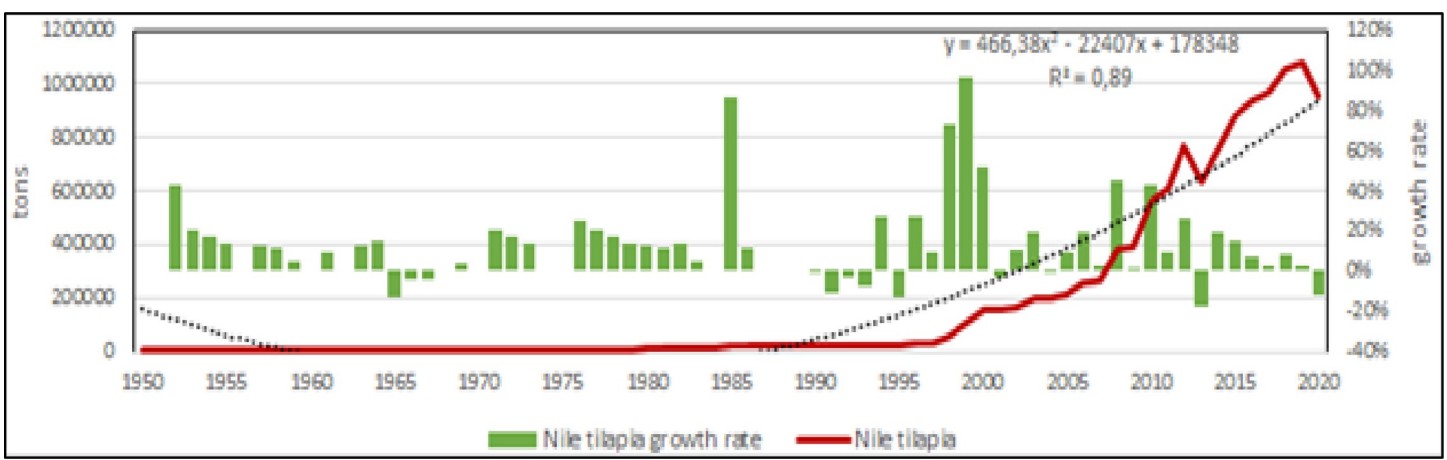

**Fig 12. Aquaculture productions of Nile tilapia.** Dotted lines: polynomial regression lines, with relative equations and regression coefficients.

and rainbow trout. That period corresponds with the diffusion of rainbow trout farming in northern countries. Nile tilapia has emerged in the region after 2000 and partially European seabass and gilthead seabream. Third period can be considered the reaching of an anthropogenic climax stage (Fig 18).

It is certainly evident the effect of technology diffusion for the progress of European sea bass and gilthead sea bream [2, 5, 29, 56–59], but the extraordinary production of Nile tilapia and mullets in Egypt indicates at the same time that technological progress are not the only possible explanations but also socioeconomic and political decisions profoundly influence aquaculture diffusion.

**Diversity.** Diversification of aquaculture is a primary goal for the future of Mediterranean productions [9, 60]. Consumer's expectation is driven by capture fishery diversification both in coastal than in inland areas [61–63]. In 2017, 83 species have been farmed, but not all the species play same role. However, the study of diversity cannot be merely considered the total number of farmed species (richness of species), as several minor farmed species are transient

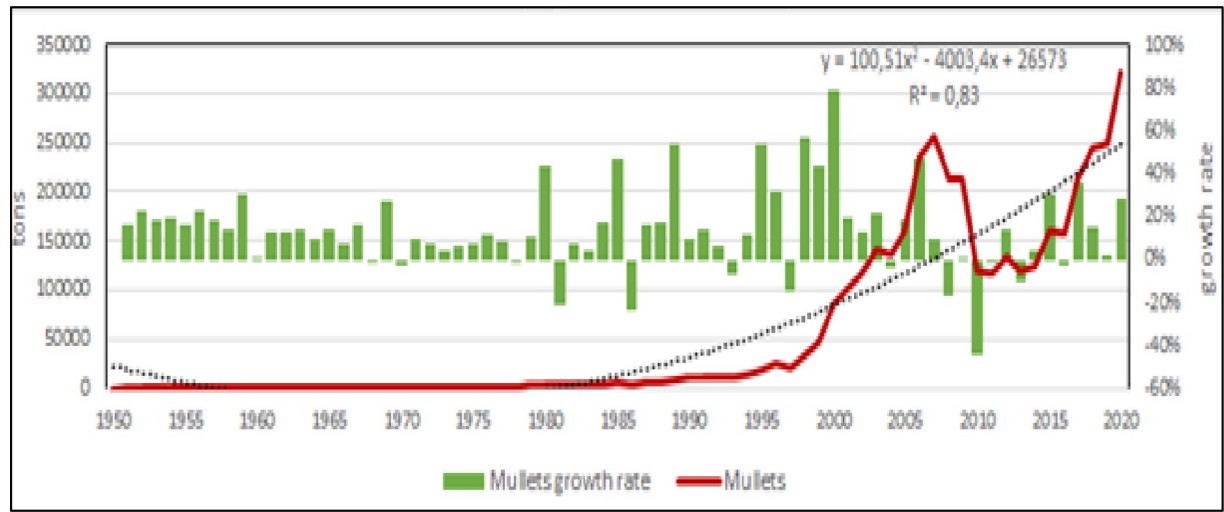

**Fig 13. Aquaculture productions of mullets.** Dotted lines: polynomial regression lines, with relative equations and regression coefficients.

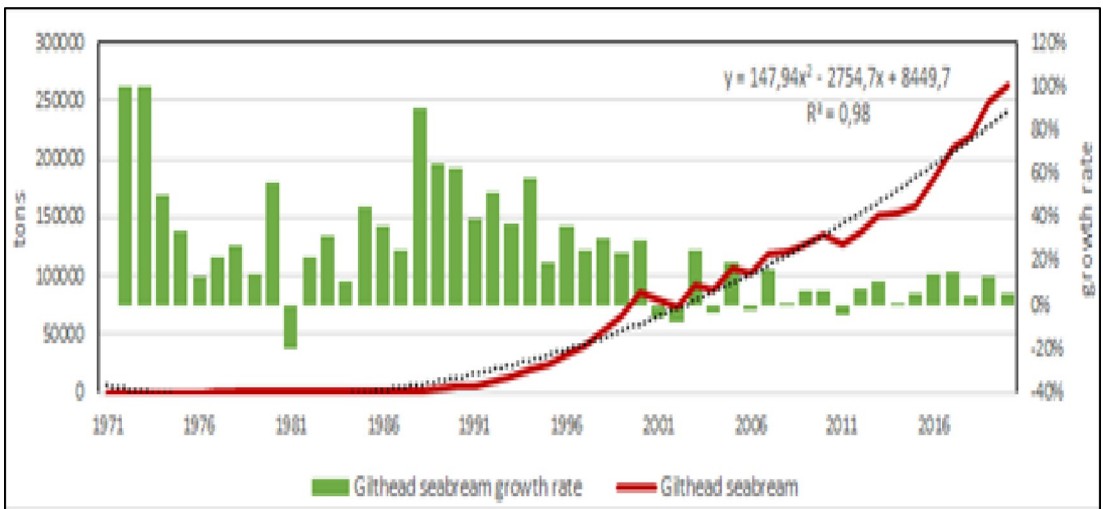

**Fig 14. Aquaculture productions of gilthead seabream.** Dotted lines: polynomial regression lines, with relative equations and regression coefficients.

both temporally than geographically. A useful analysis of aquaculture diversity must be based on the functional role of farmed species, not a mere enumeration of species. This approach makes evident two aspects of aquaculture diversity: dominance of few species and redundancy of minor ones. These concepts are very well known in applied ecology [64] and in marine resources management [8], consequently three main diversity components have been considered: dominant, relevant and minor species. The Pareto principle (or so-called 80/20 rule) states that, for many events, roughly 80% of the effects come from 20% of the causes. The Pareto distribution is very common, for instance in economics (income distribution) or in sociology (the 80% of crimes are committed by 20% of criminals.) or in fish parasite

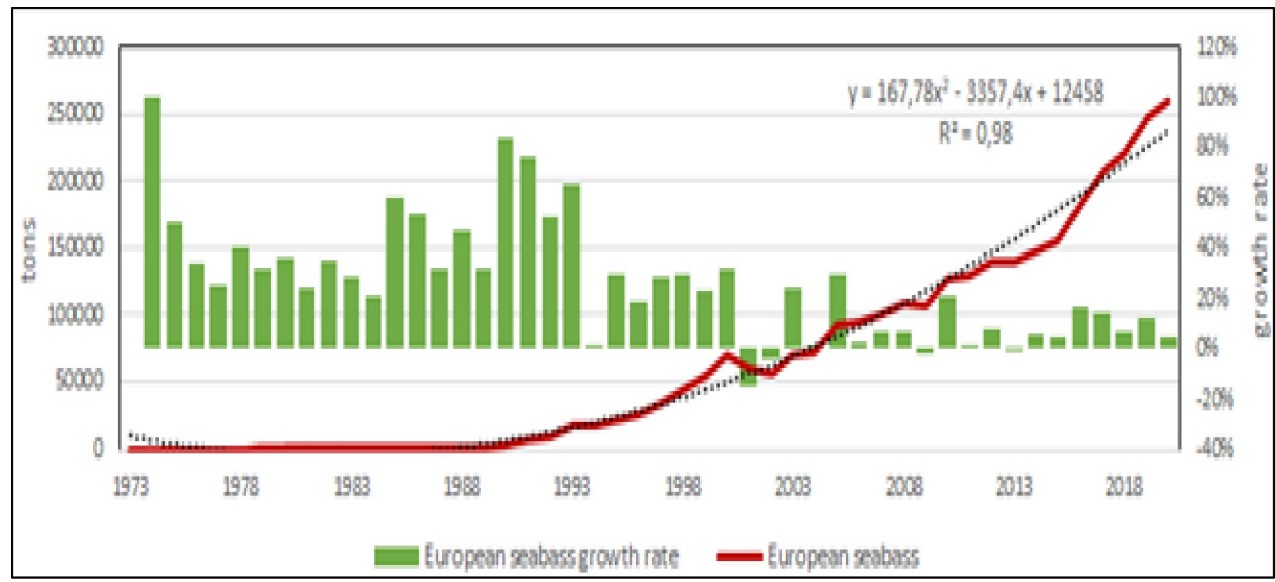

**Fig 15. Aquaculture productions of European sea bass.** Dotted lines: polynomial regression lines, with relative equations and regression coefficients.

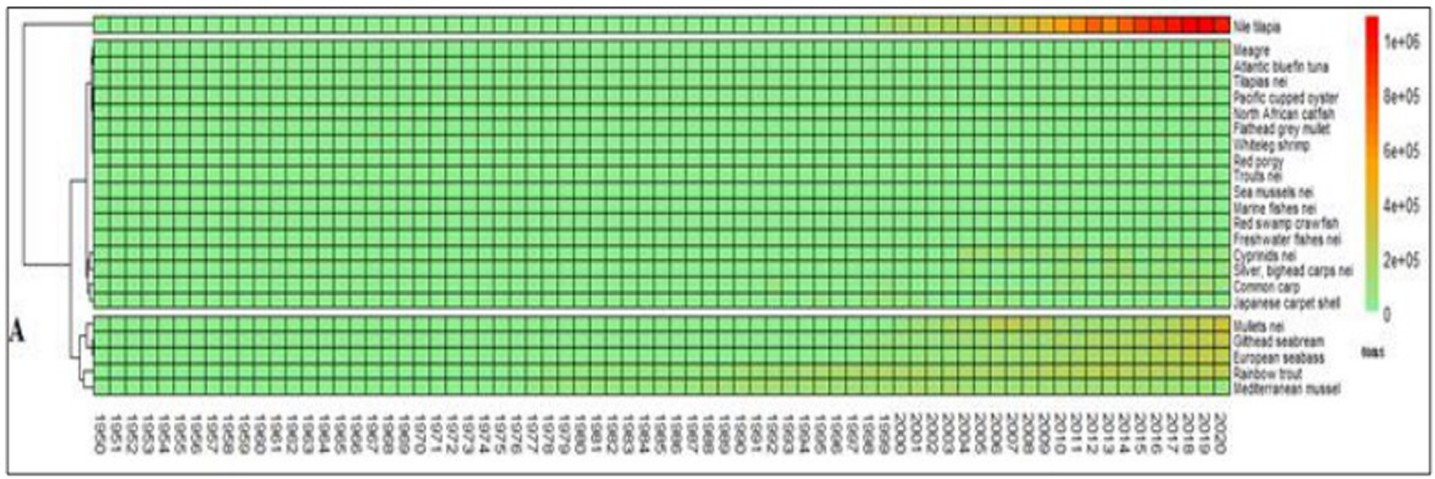

**Fig 16. Heatmaps of main aquaculture species in the Mediterranean: Original data.**

distribution and other cases. It can be useful to introduce it in the study of aquaculture diversity. For the application of Pareto index to the study of aquaculture diversity, minor species, *i. e.* those accounting for of less of 1% of annual production, have been eliminated. Only those species the contribute for the 99% of annual aquaculture production have been considered. According to Pareto principle, these species have been divided in two groups: those that produce the 80% and the resting species that produce the 20% of annual production. As example, in 2017, on a total of 83 farmed species, 22 species produced the 99% of annual production (thereafter called "Relevant species"), other 61 species produced less than 1% of annual production. Within Relevant species, only 5 (4% of total number of species, thereafter called "Main species") produced 80% of total aquaculture volume (Figs 19 and 20). These species can be considered the Pareto's "vital few" of aquaculture [27].

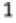
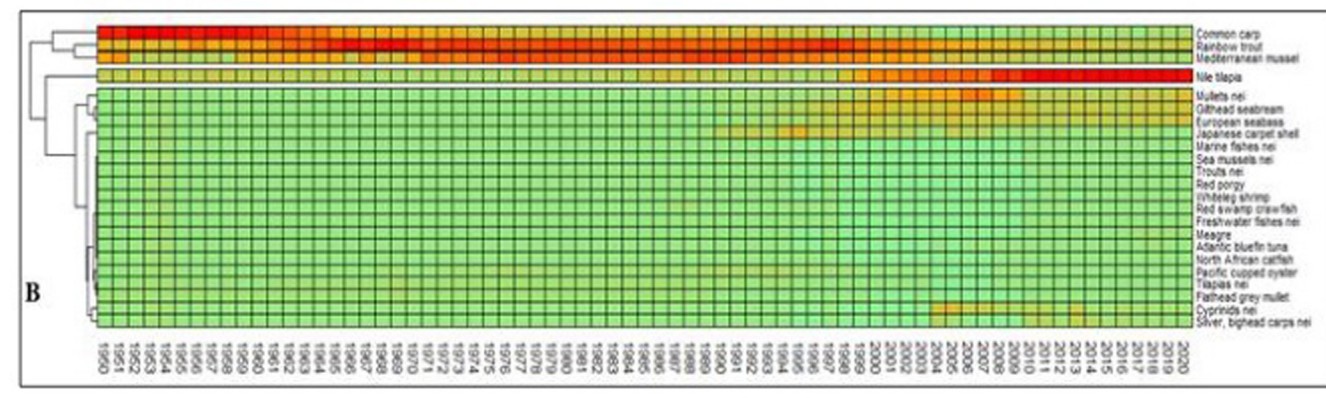

**Fig 17. Heatmaps of main aquaculture species in the Mediterranean: Scaled data.** Data scaling: $x_{scaled} = (x-\mu)/s.d.$ ($\mu$ = mean; s.d. = standard deviation).

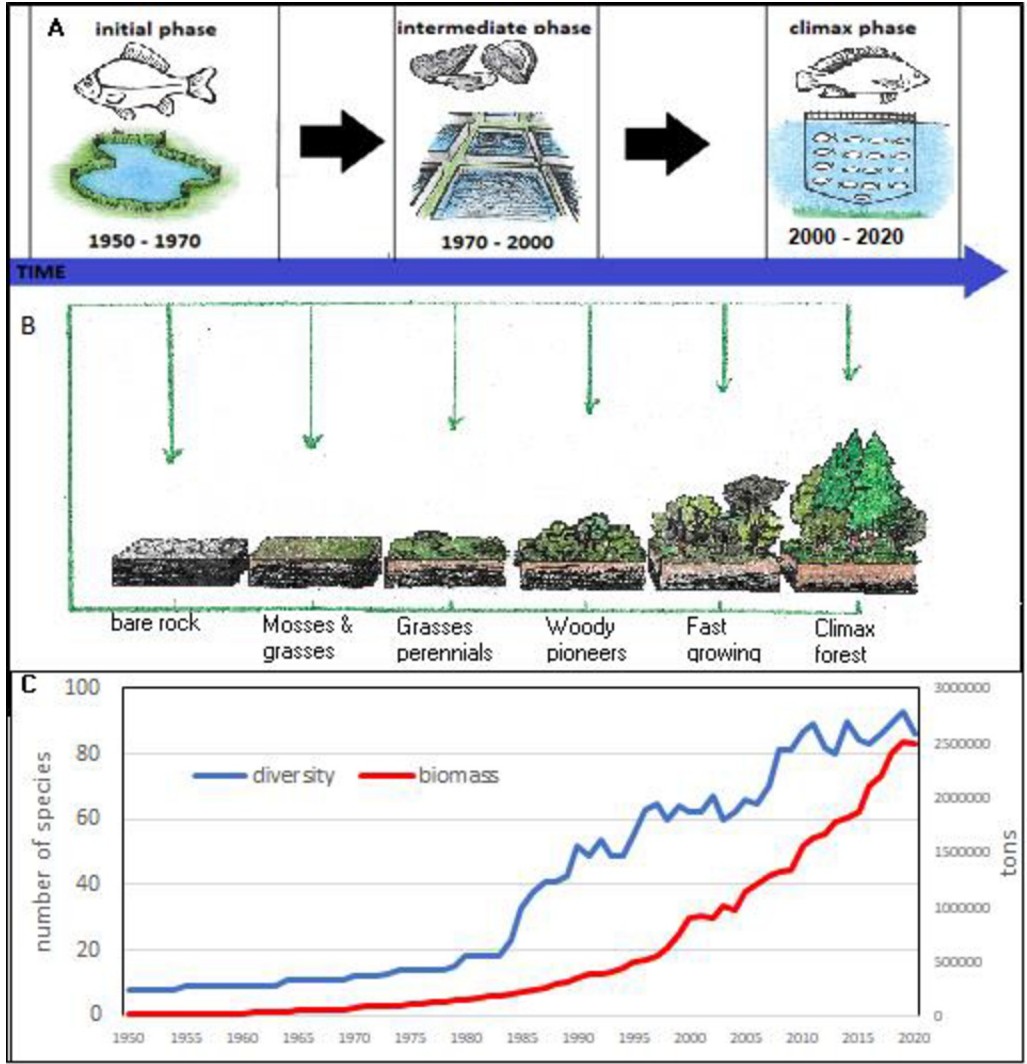

**Fig 18.** Comparison between succession of farmed species in the Mediterranean in 3 stages (A and C) and a forest succession in 6 stages (B).

The Pareto index, *i.e.*, the ratio between relevant and minor species, shows that there are two main phases in the evolution of aquaculture in the Mediterranean region (Figs 19 and 20); before the 1984 the number of species producing the 80% of total volume was the 30%, while in a second phase an increase of dominance of few species has showed that the 20% of species has produced the 80% of volume. Finally, it is possible to foresee that in the future the 80% of aquaculture volume will be provided by 4 or 5 species and the 99% of volume by 22 to 25 species (Fig 20).

The gradual progression of aquaculture diversity is confirmed by the application of Shannon index that is much less variable respect number of species (Fig 21). Shannon index indicates that diversity has reached a stationary state phase after 2010, confirming Pareto index analysis. Shannon diversity is a measure of functional diversity, as it includes not only the number of species but also their occurrence that in this case has been measured as volume of production of each species. The stationary phase of Shannon index after 2010 in the

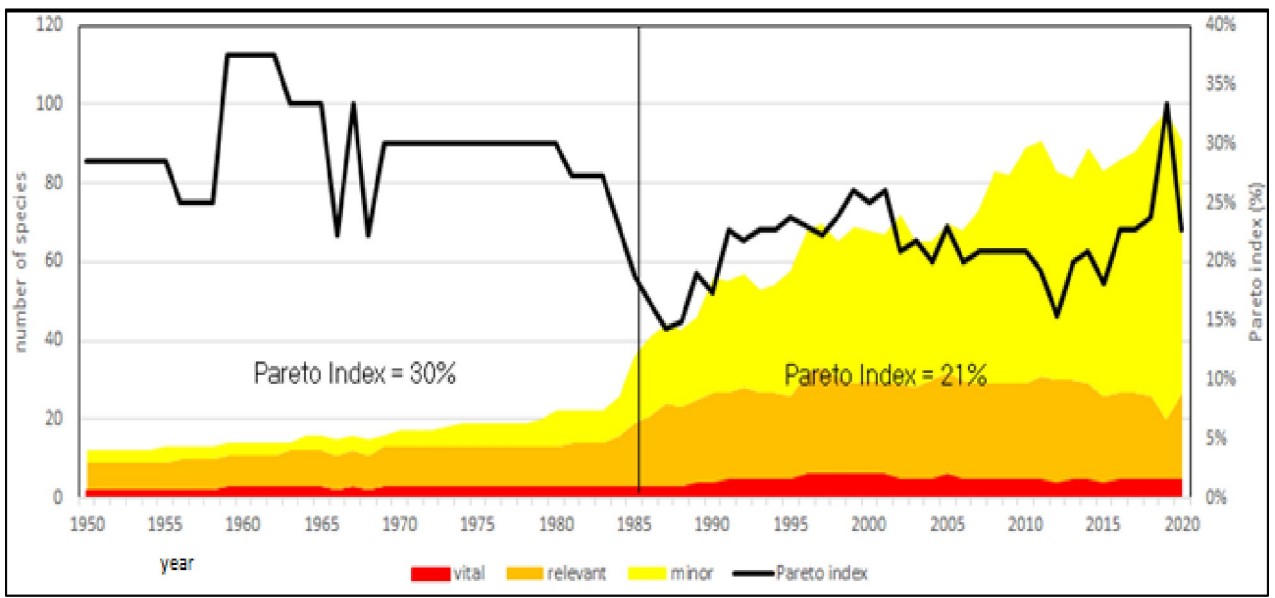

**Fig 19. Aquaculture diversity in the Mediterranean region, considering its main components: Main, relevant and minor species.** A: Pareto Index trend (ratio main/relevant species).

Mediterranean region and the small decrement after 2021 can indicate an instable state related to the extraordinary growth of aquaculture production caused by few species.

## The aquaculture geography of Mediterranean

Four aspects characterize the geographical expansion of aquaculture: volumes of production, number of countries where aquaculture is present (diffusiveness), number of farmed species per country (diversity) and number of years of aquaculture farming per country (persistence).

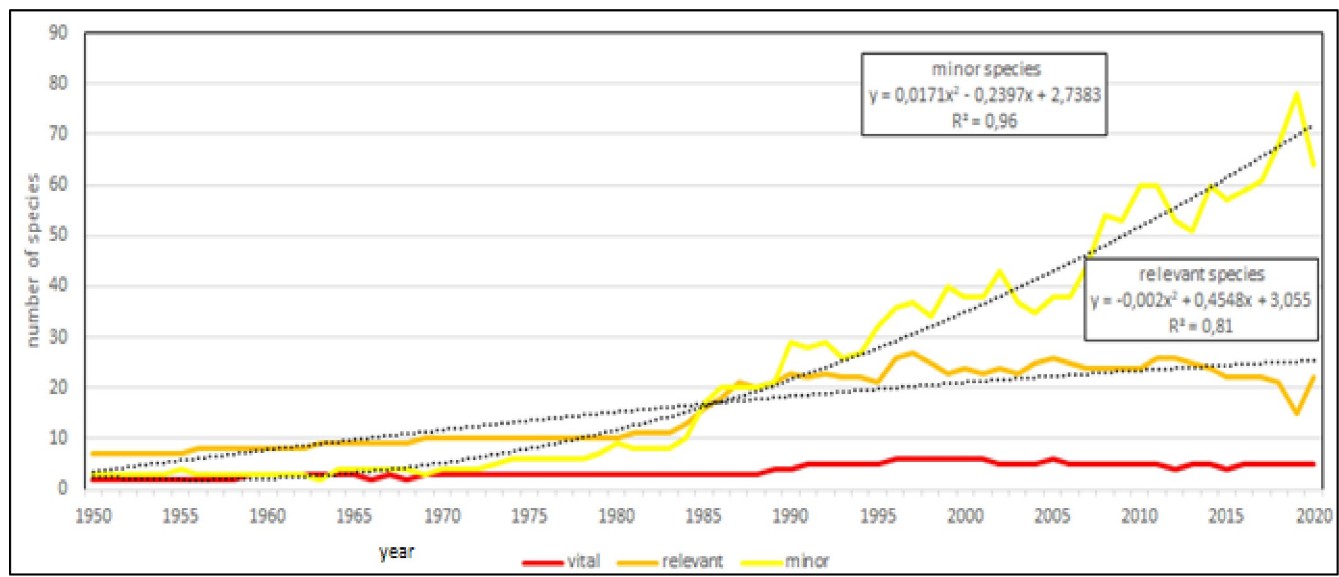

**Fig 20. Aquaculture diversity in the Mediterranean region, considering its main components: Main, relevant and minor species: Evolution of different components.** Dotted lines: polynomial regression lines, with relative equations and regression coefficients.

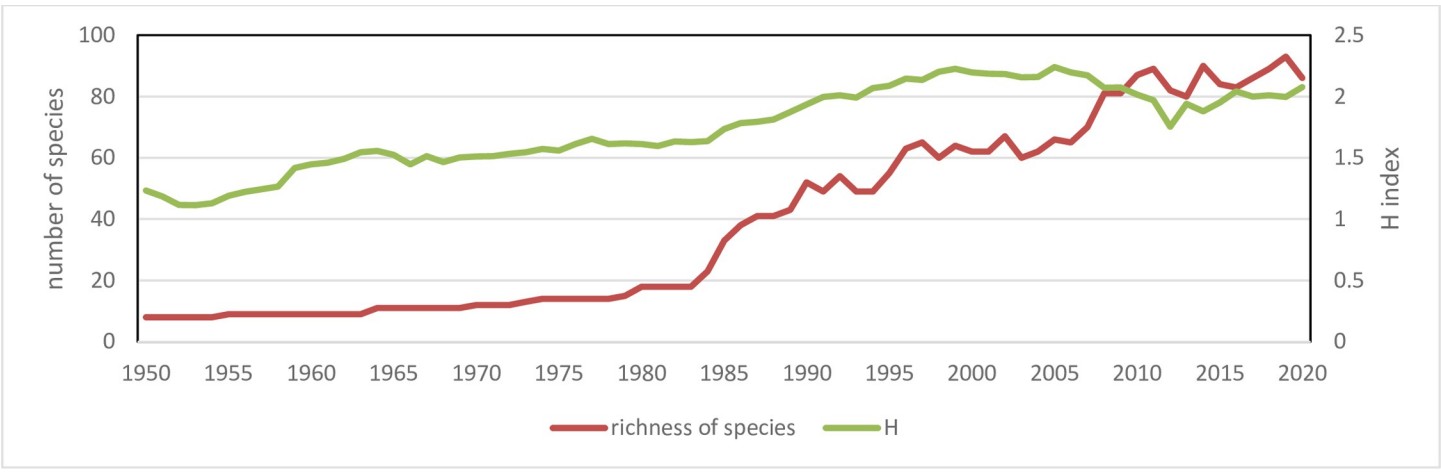

**Fig 21. Aquaculture diversity in the Mediterranean region.** Shannon Index and species richness.

**Volumes.** The expansion of aquaculture in the Mediterranean has been caracterised by a progressive movement in south East direction explanable by the diffusion of marine aquaculture technologies in Greece and Turkey [1, 9, 65–67] and by the tremendous growth of tilapia farming in Egypt [15].

Based on aquaculture volumes, Mediterranean region shows 3 main groups of countries: Egypt; north Mediterranean plus Greece and Turkey; a third group of countries comprising Balkans, South Mediterranean and middle East countries, both considering cumulative than last 5 years productions (Fig 22). Egypt prominent position is essentially determined by the diffusion of Nile tilapia, extensively farmed [16, 25, 29, 43, 68, 69]. In Egypt, almost 89% of aquaculture is produced in ponds [5], it has more than 2450 km of coastline, as well as 8700 km$^2$ of inland water and most aquaculture of its production is obtained in the Nile delta region, with tilapias, mullets and carps making up more than 97% of total production in 2007 [5, 67].

The second region is mainly characterized by those countries where European sea bass and gilthead seabream farming is well developed [65, 70]. These countries are also those with a medium/high technology level, and aquaculture products are economically relevant as exportation products [9, 71–73]. The third region is composed by countries with low volumes of production, a heterogeneous group of countries containing both huge north African countries then East Europeans ones, these countries potentially represent next area of aquaculture expansion.

At country level, aquaculture development is primarily influenced by governance infrastructure [23], local food consumption traditions and geographical differences that favors different aquaculture species. The diffusion of aquaculture is largely influenced by socio/political and economic factors [74, 75]. To give a partial description of these factors, the World Bank data (World Bank national accounts data, and OECD National Accounts data files; https://datacatalog.worldbank.org/public-licenses#cc-by, Last Updated: 23/11/2021), have been correlated with aquaculture productions, considering the period corresponding to aquaculture value data, from 1984 to 2015 (Figs 23–25).

Considering the 3 aquaculture zones previously observed (Fig 22), there is a strong correlation between aquaculture value and Gross Domestic Product (GDP) pro capita, thus indicating that aquaculture plays an important role in these economies, particularly in Egypt (Figs 23–25). Fish consumption (Figs 26–28) is also correlated with aquaculture production and considering that capture fishery has reached its maximum at the end of 20$^{th}$ century (Fig 4), the

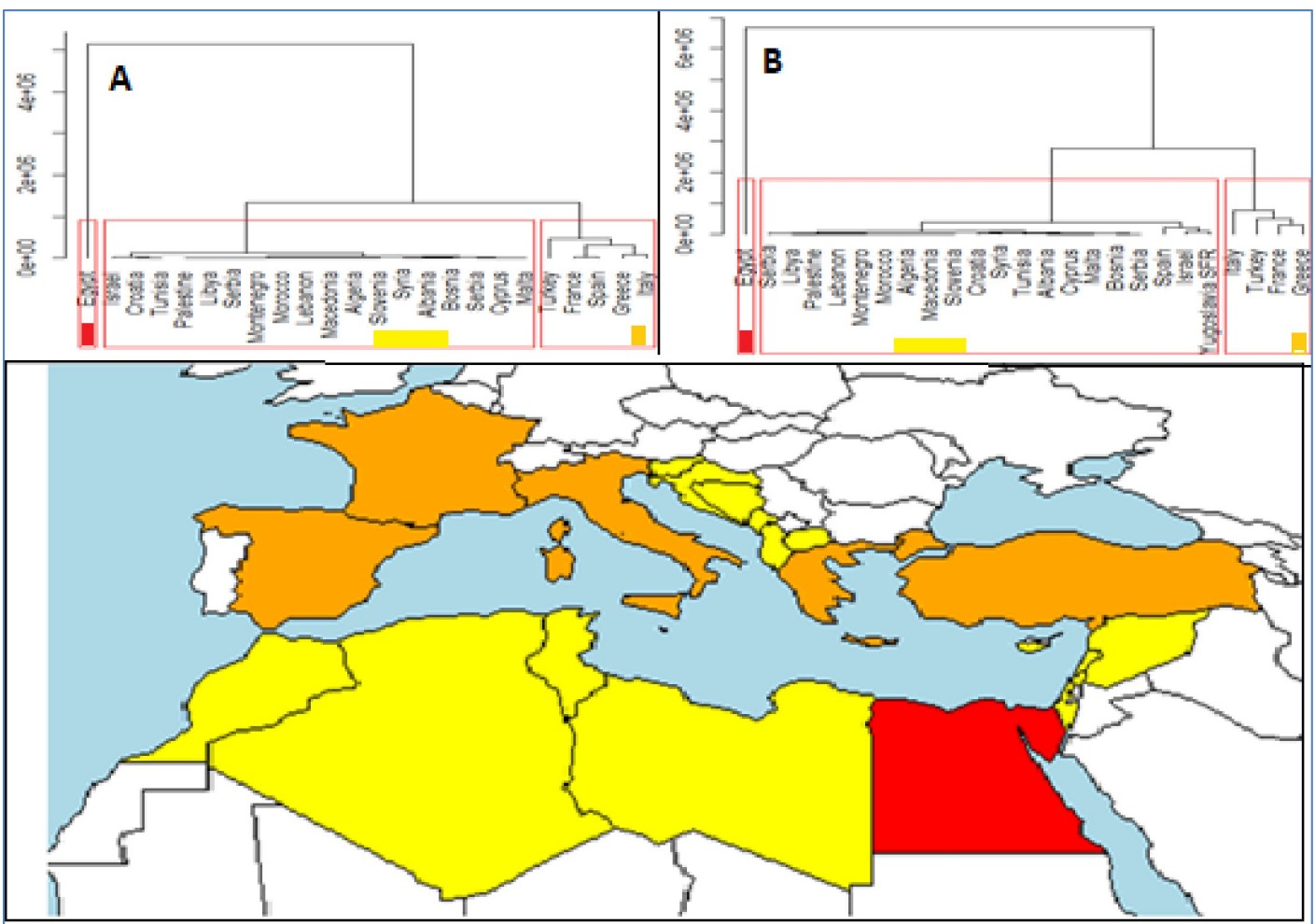

**Fig 22. Geographical map of aquaculture of Mediterranean aquaculture\*.** (A, cluster obtained considering the entire period; B, cluster obtained considering las 5 years). \* Data from parts of the countries on Atlantic Ocean or Red sea coasts (France, Morocco, Spain and Egypt) have been excluded by geographical analysis.

future fish consumption in the region will be substantially sustained by aquaculture productions. As example, the estimated model for fish consumption in Egypt ($y = 0,58x + 3.46$; $r = 0,95$; y = year; x = fish consumption *pro capita*), indicate that in 2030 fish consumption will be about 31 kg/person/year. These consumptions certainly indicate a great expansion of seafood market which necessarily will rely on aquaculture productions.

**Country diversity.** Geographical diffusion of aquaculture is certainly affected by the technological level of countries [76]. In the Northwestern countries of Mediterranean there have been farmed more species (Figs 29 and 30), this is an indication of higher capacity of innovation and also depends from the historical origin of aquaculture in North of the region, in the European countries [77]. However, shifting from species richness to Shannon index (Fig 29) the initial differences between northern and southern countries are tempered. H index is a more suitable indication of diversity [74] and the relative map (Fig 29) indicates that in Morocco, Algeria, Tunisia and Israel, aquaculture depends on a few species, but reared in comparable volumes between species. The unexpected inclusion of some north African countries in the group of countries with higher values of H index indicates a more balanced development of aquaculture.

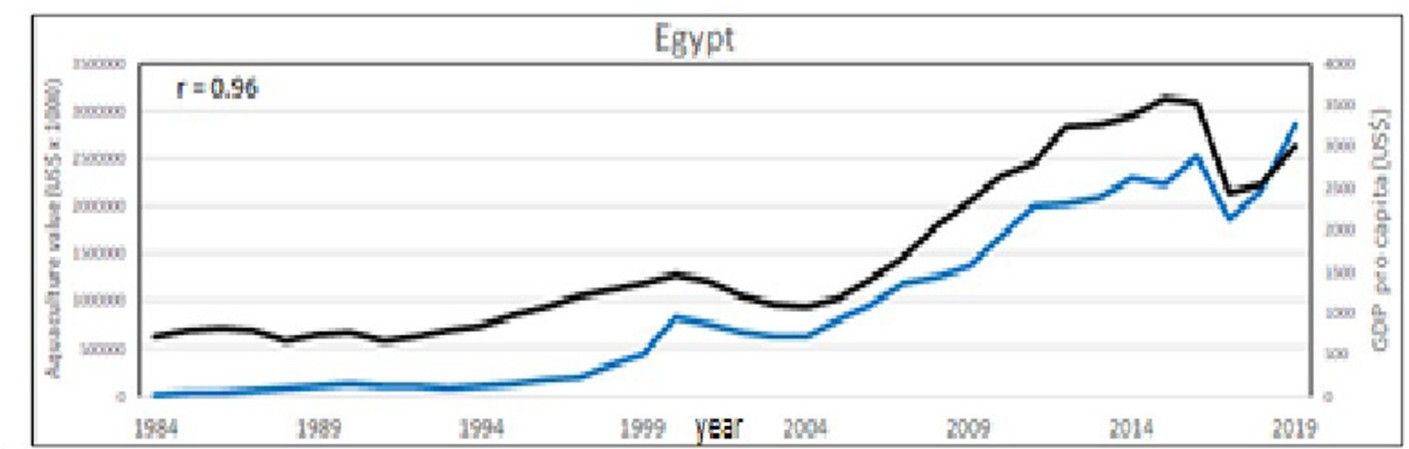

**Fig 23. Aquaculture value (blue line) and GDP *pro capita* (black line) in Egypt.** In the corner, on the left, the Pearson correlation coefficient (r).

**Persistence.** The past history of farming can be certainly considered a decisive factor affecting the aquaculture development, as shown by the world dominance of China aquaculture that is the oldest and largest producer country in the planet (Fig 31). The analysis of persistence of aquaculture in the Mediterranean primarily shows that aquaculture is a rooted activity, in particular in the northern coast and in the most productive countries aquaculture has started in a period comprised by 55 and 66 years ago, while in almost all the southern Mediterranean countries, aquaculture is a more recent activity. An exception is in South western countries is Israel, where commercial marine aquaculture began in the late 1980s [78].

As previously observed in the study of the farmed species (Figs 19 and 20), the utilize of Pareto principle can facilitate the interpretation of aquaculture expansion in the Mediterranean, thus including the volumes of production. Therefore, dominant countries have been considered those that annually produced the 80% of total production of aquaculture in the

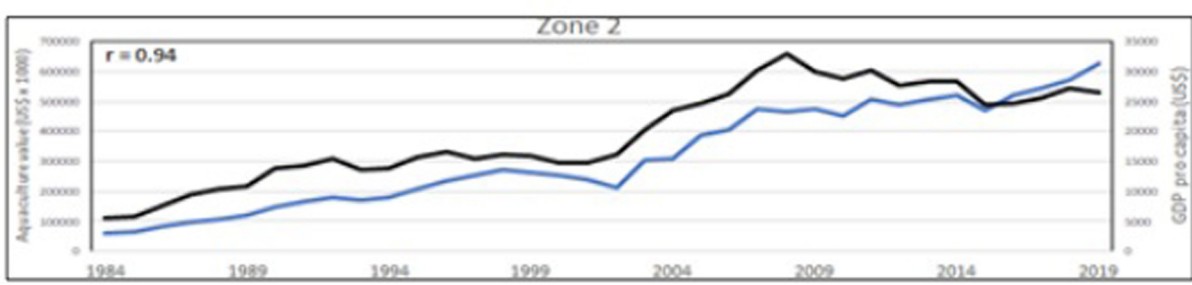

**Fig 24. Aquaculture value (blue line) and GDP *pro capita* (black line) in the zone 2 (Fig 22).** In the corner, on the left, the Pearson correlation coefficient (r).

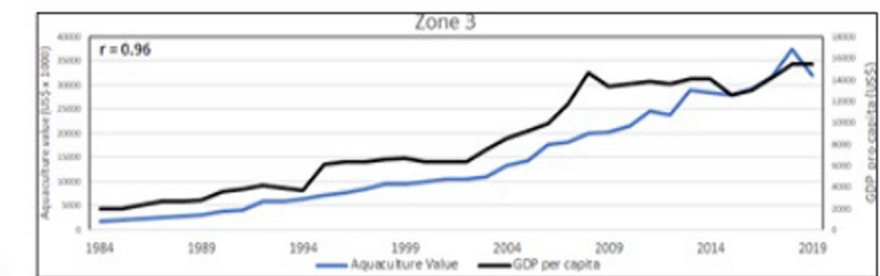

**Fig 25. Aquaculture value (blue line) and GDP *pro capita* (black line) in the zone 3 (Fig 22).** In the corner, on the left, the Pearson correlation coefficient (r).

region (Fig 32). This simple analysis shows that during almost the entire history of Mediterranean aquaculture, only 2 or 3 countries have produced the 80% of annual volume.

More interestingly, the evolution of ratio between main countries and total number, the Pareto index (Fig 32), shows that it is about 23% for the entire considered period, even if in the first years there are some oscillation presumably caused by low number of countries producing the 80% of total production. Interestingly, the geographical expansion is dominated by few countries but not the necessarily the same ones along the period considered (in fact, in the beginning first countries were France and Italy and successively Egypt). Italy has progressively

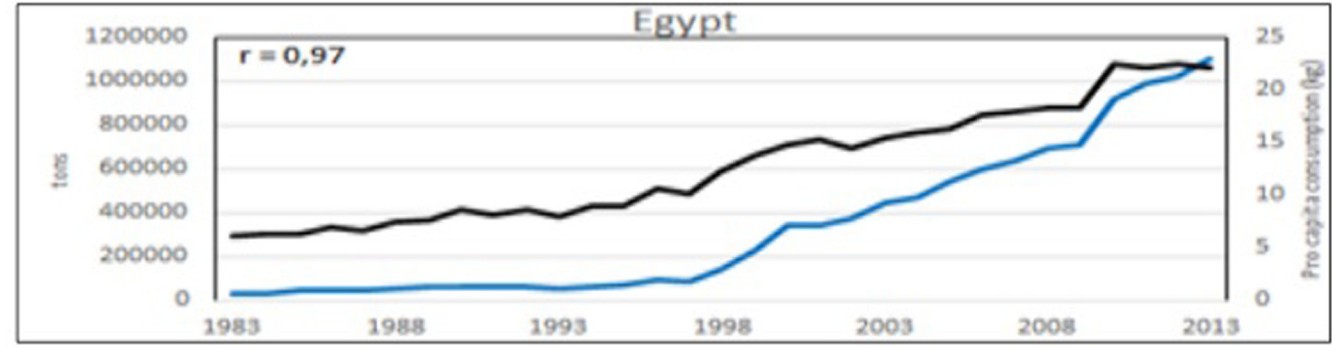

**Fig 26. Aquaculture production (blue line) and sea food\* consumption *pro capita* (kg/year) (black line) in Egypt.** In the corner, on the left, the Pearson correlation coefficient (r). (\*sea food includes: cephalopods, crustaceans, demersal fish, fish body oil, fish liver oil, freshwater fish, marine fish, other molluscs, other pelagic fish, according all the items contained in the database: https://www.fao.org/faostat/en/#data/CL).

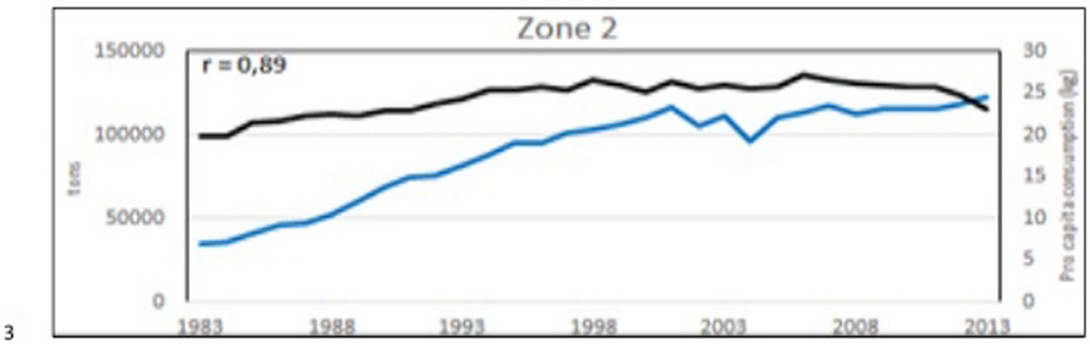

**Fig 27. Aquaculture production (blue line) and sea food\* consumption *pro capita* (kg/year) (black line) in the zone 2 (Fig 22).** In the corner, on the left, the Pearson correlation coefficient (r). (\*sea food includes: cephalopods, crustaceans, demersal fish, fish body oil, fish liver oil, freshwater fish, marine fish, other molluscs, other pelagic fish, according all the items contained in the database: https://www.fao.org/faostat/en/#data/CL).

lost its prominent position in relation with increasing of volumes caused by the diffusion of aquaculture technologies, thus confirming the role of tecnologies in the region [58, 76, 79], together with political and regulatory changes occurred in the region as measures of

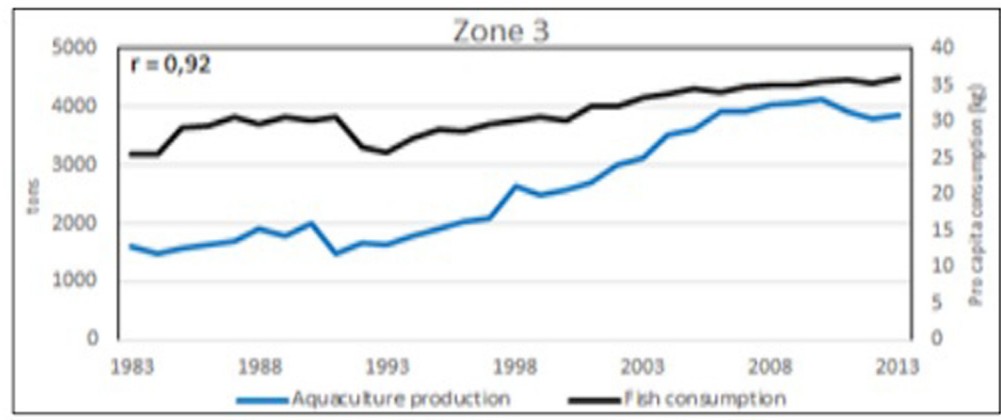

**Fig 28. Aquaculture production (blue line) and sea food\* consumption *pro capita* (kg/year) (black line) in the zone 3 (Fig 22).** In the corner, on the left, the Pearson correlation coefficient (r). (\*sea food includes: cephalopods, crustaceans, demersal fish, fish body oil, fish liver oil, freshwater fish, marine fish, other molluscs, other pelagic fish, according all the items contained in the database: https://www.fao.org/faostat/en/#data/CL).

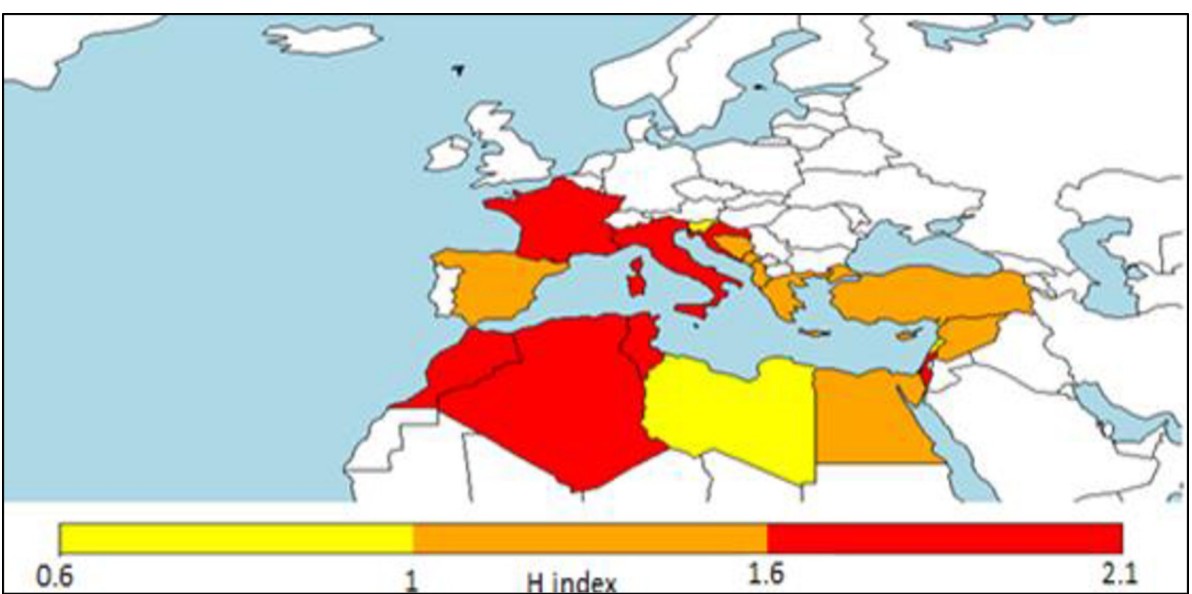

**Fig 29. Maximum diversity map (Shannon index) based on maximum values out of 67 years in each country.**

aquaculture promotion al local level [78, 80]. Pareto indexes calculated on countries and species diversity (Figs 19, 20 and 32) are comparable, thus indicating an interesting convergence.

**Relative productivity: Where the last becomes first.** The relative productivity analysis shows that Malta is the first country (Table 1). Even if the production/surface ratio is a rough measure of aquaculture potentiality, these values indicate that aquaculture has a tremendous potentiality in the Mediterranean [76]. Malta is a small island and for its peculiar geo/political conditions [78, 80] it cannot be considered as a suitable model of aquaculture development, in fact the recent aquaculture growth has been also locally criticized for the negative effects on

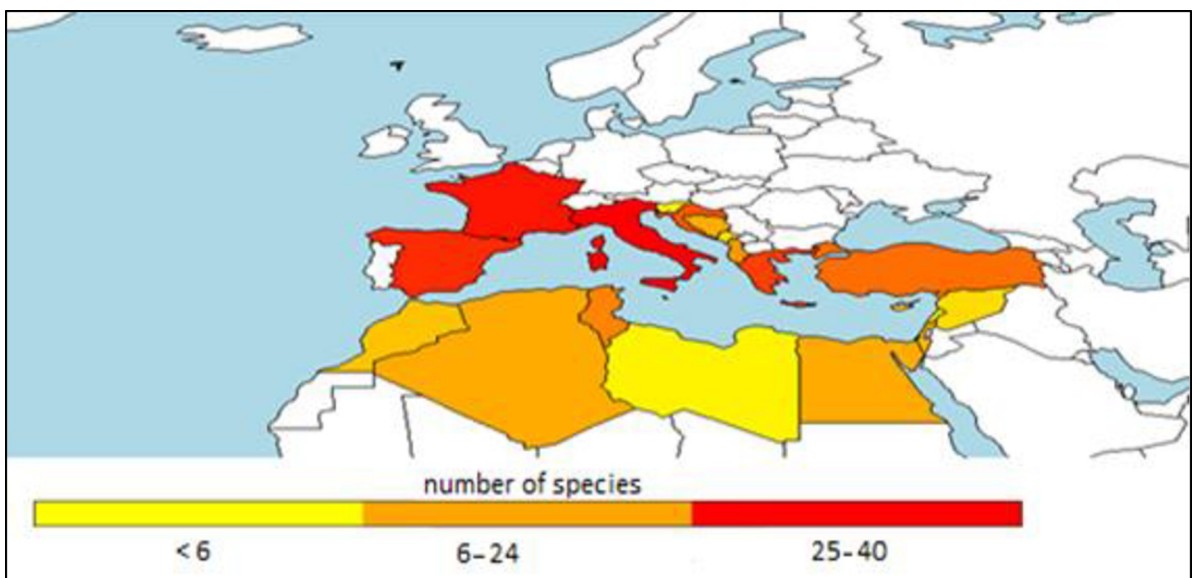

**Fig 30. Maximum diversity map (number of species) based on maximum values out of 67 years in each country.**

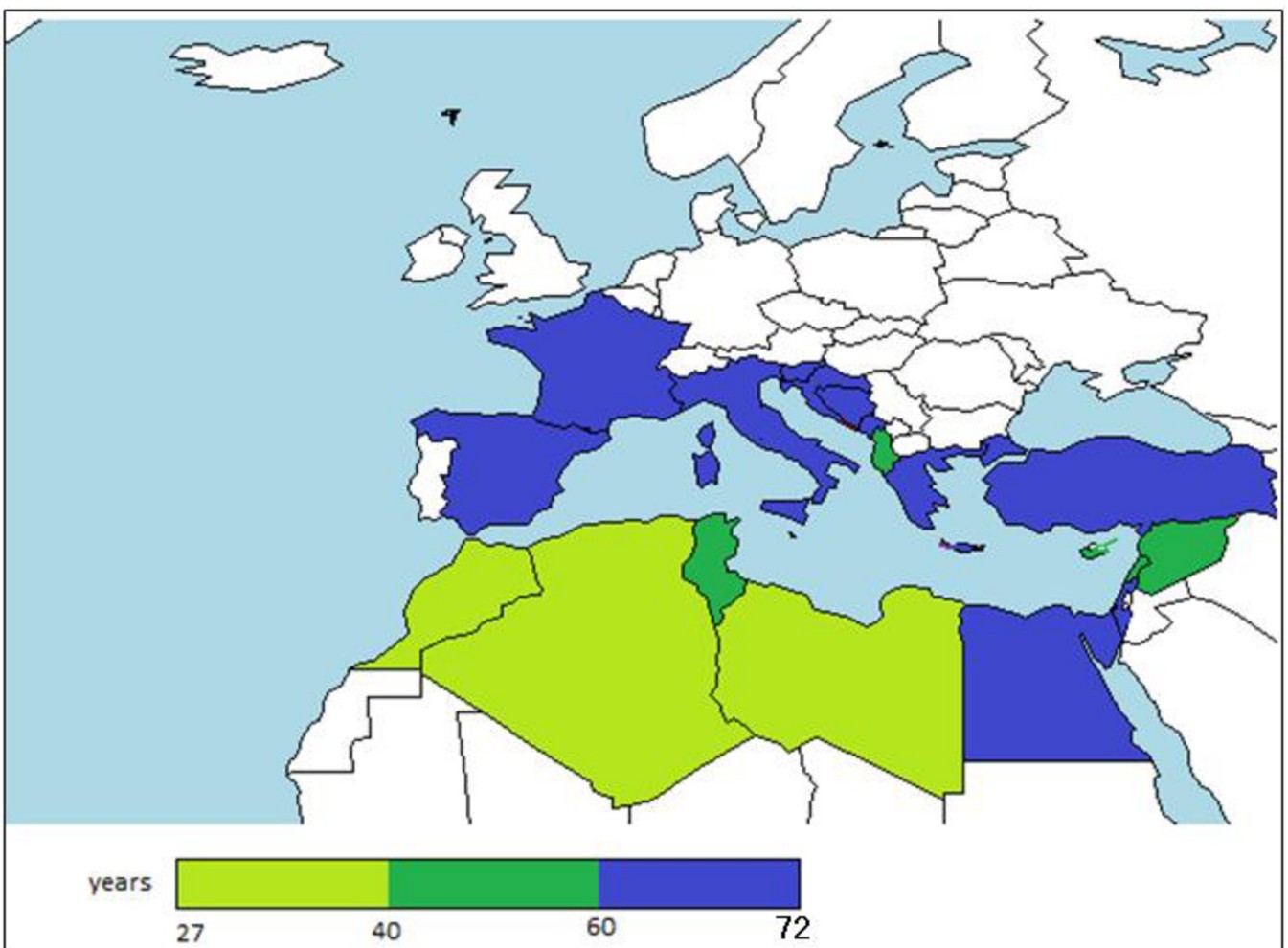

**Fig 31. Map aquaculture persistence (years of farming in each country) *countries corresponding to ex-Yugoslavia have been counted considering the historic change of national boundaries.**

fishing enterprises [81]. Therefore, even excluding Malta, the examples of Israel, Egypt and Greece productivity (average 5 tons/km$^2$/year) can be taken as sustainable target for other countries development. In Malta, Egypt and Israel capture fishery is constant or declining, as noticed in previous studies [69, 82], but aquaculture production is rapidly growing, in particular in Egypt [15] where the growth is exponential and in 2030 aquaculture productions are expected to reach more than 2 Mln tons. In Egyptian aquaculture, mullets farming has progressively acquired relevance in the late 80ities, exerting a kind of competition by exclusion with carp (Figs 33 and 34). In absolute terms Malta and Israel show more variations in the ratio aquaculture /capture fishery, as effect of minor volumes, but what is really interesting is the extraordinary growth of aquaculture respect capture fishery and a formidable shift from fishery to aquaculture even in Egypt. In fact, in Israel and Malta the production of aquaculture in 2015 was even 8 times more that of capture fishery (Figs 35–37).

Therefore, Malta confirms its extraordinary relative growth not only respect to its size but also respect to capture fishery. The reasons of this outstanding development of aquaculture must be searched on local economic and socio/political measures on aquaculture development [22, 57, 59, 83]. With the entry of Malta into the EU in 2004, the fisheries management zone

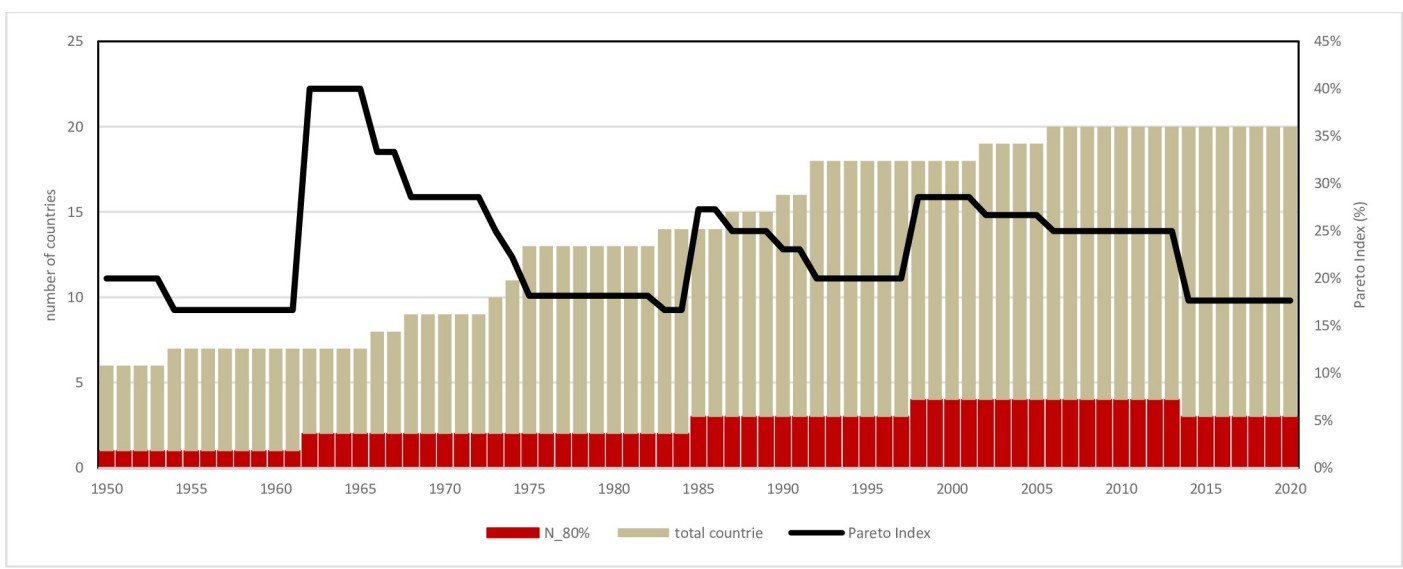

**Fig 32. Geographical expansion of aquaculture in the Mediterranean region.** In red the number of countries producing the 80% of annual production. Pareto index (%) = (number of countries producing the 80% of annual production/total countries aquaculture producing).

around Malta by an EU Council regulation covered 11980 km$^2$, which is over 38 times its terrestrial extent [79]. National and EU financing of infrastructures in Malta have largely contributed to the technological development of existing aquaculture farms as well as to the start-up

**Table 1. Relative productivity (In the last 5 years, from 2013 to 2017) and persistence of aquaculture.**

| Country | Volumes (tons/km$^2$) | Value (Mln€/ km$^2$) | Persistence (years) |
|---|---|---|---|
| Malta | 92,6 | 211,6 | 27 |
| Egypt | 6,2 | 1,8 | 67 |
| Israel | 4,5 | 4,1 | 67 |
| Greece | 4,4 | 4,4 | 67 |
| Cyprus | 3,3 | 4,3 | 44 |
| Italy | 2,5 | 1,6 | 67 |
| Turkey | 1,6 | 1,2 | 64 |
| Croatia | 1,3 | 1,6 | 26* |
| Albania | 0,7 | 0,1 | 44 |
| France | 0,5 | 0,4 | 67 |
| Spain | 0,5 | 0,5 | 63 |
| Tunisia | 0,5 | 0,2 | 41 |
| Lebanon | 0,5 | 0,2 | 42 |
| Bosnia | 0,4 | 0,1 | 16* |
| Slovenia | 0,4 | 0,1 | 26* |
| Montenegro | 0,3 | 0,1 | 12* |
| Syria | 0,1 | 0,1 | 47 |
| Morocco | 0,1 | 0,1 | 36 |
| Algeria | 0,1 | 0,1 | 34 |
| Libya | 0,1 | 0,1 | 31 |
| Palestine | 0,1 | 0,1 | 13* |

* in these countries, for political reasons, the persistence must be considered the maximum possible

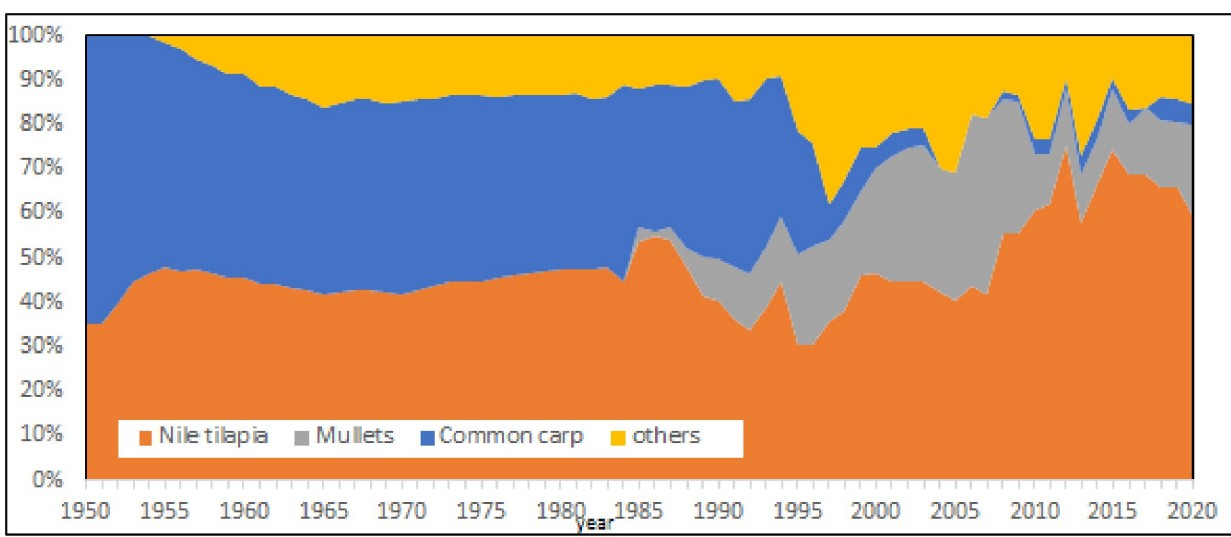

**Fig 33. Aquaculture productions in Egypt (main species) from 1950 to 2020 (annual %).**

of new ones within European countries [5] and other areas of the world [23]. In Israel and Malta aquaculture has been supported by relevant state economic investments and initiatives, [78, 80] as happens developed countries such as Japan or New Zealand [84–86]. The diffusion of modern technologies and a modern legal system of aquaculture governance [76, 87] is determinant in Israel [36, 45, 58, 78]. Moreover, there is another indirect evidence that supports this hypothesis: Greece, with a coastal length of 13673 km and hundreds of islands [60], is overpassed in relative terms by Malta and Israel that is covered by desert for 60% of its surface and a coastline of only 273 km.

**Aquaculture indexes.** To further investigate the main factors affecting aquaculture, countries can be divided into 3 groups with high, medium and low production and correlated with

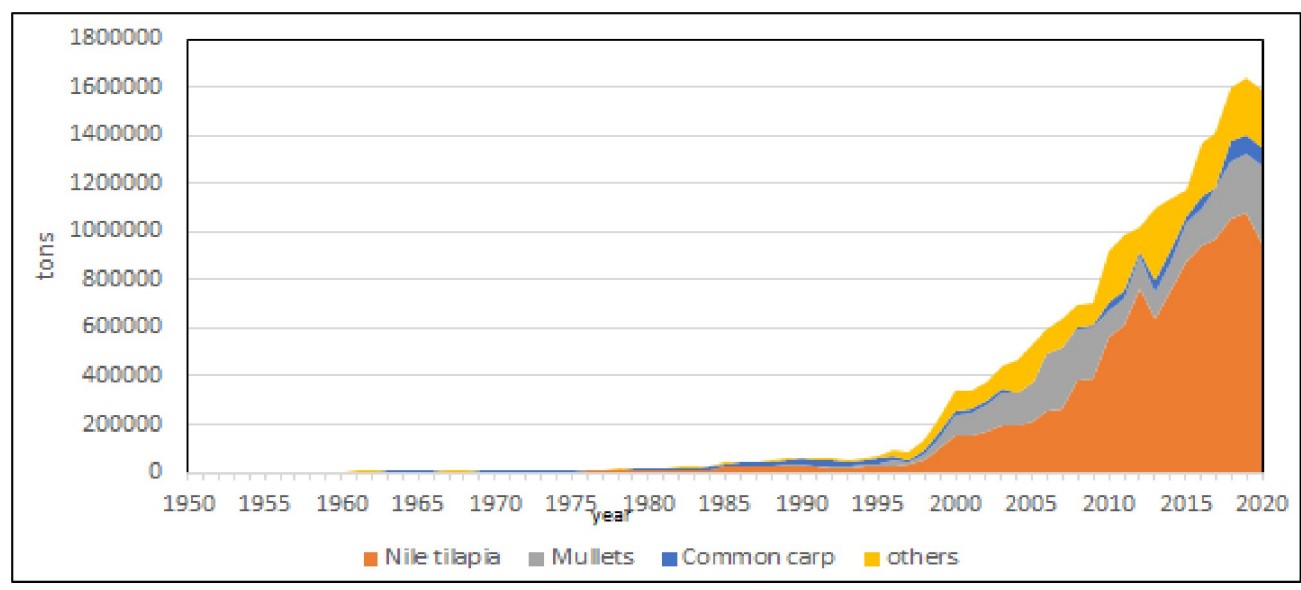

**Fig 34. Aquaculture productions in Egypt (main species) from 1950 to 2020.**

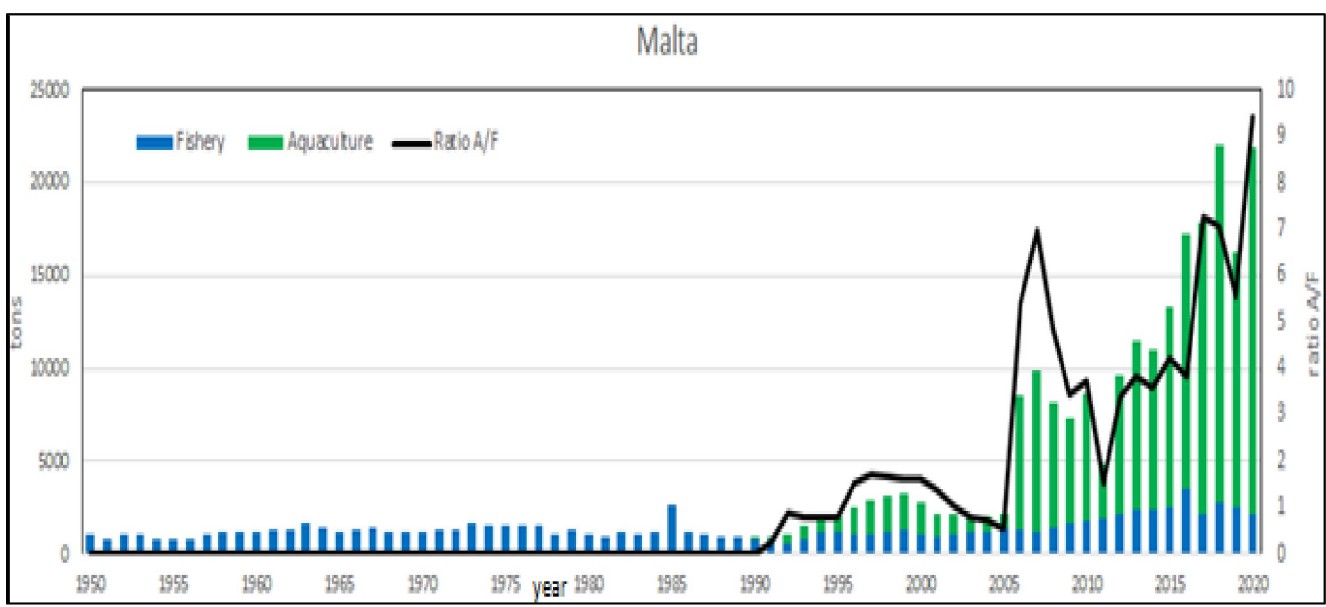

**Fig 35. Aquaculture and fishery production in Malta from 1950 to 2020.** Black lines are the aquaculture/fishery volume ratio.

socio-economic indexes (Figs 38 and 39). This analysis confirms that most of Mediterranean countries have low production and the outlier position of Egypt (Figs 38 and 39). The absence of correlation with main geographical parameters indirectly suggests the effect of external factors to aquaculture, as economic and socio-political factors. The only noticeable correlation is between persistence and diversity which indicates that countries with older tradition in aquaculture have a higher propension for innovation (Fig 38).

The exclusion of Egypt (Fig 39) makes clearer that aquaculture persistence and diversity have influenced the success of countries, in fact Turkey, Italy and Greece, (the intermediate

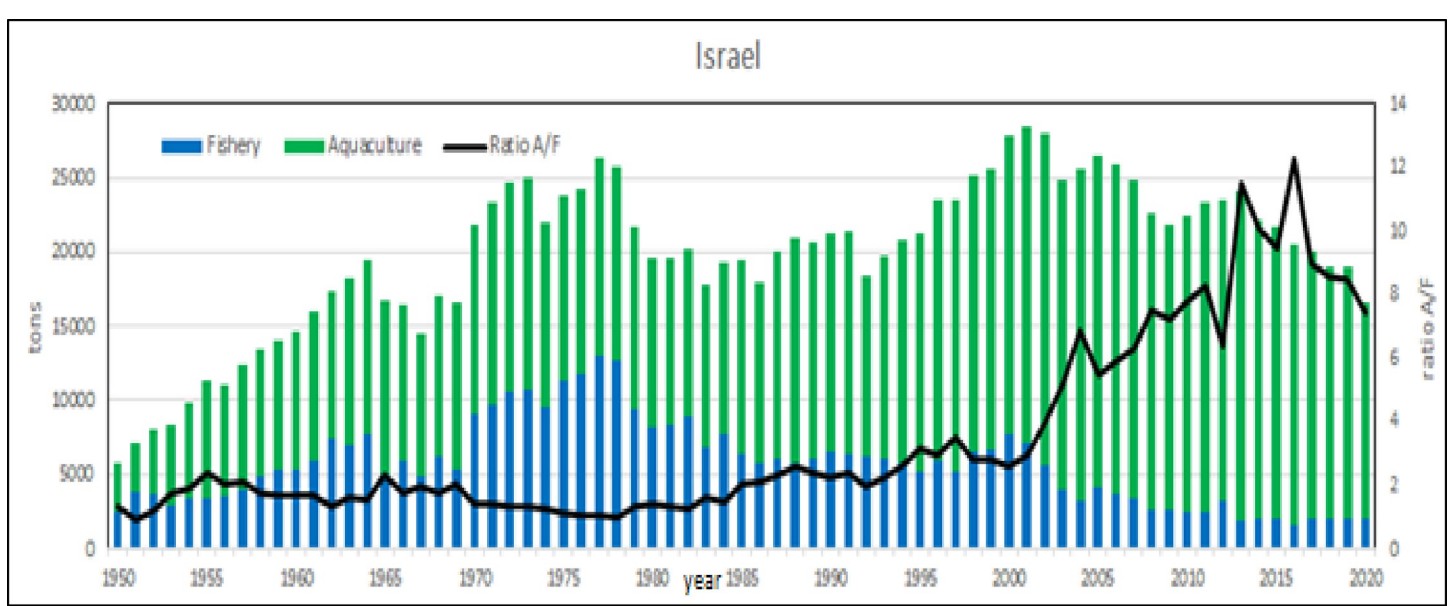

**Fig 36. Aquaculture and fishery production in Israel from 1950 to 2020.** Black lines are the aquaculture/fishery volume ratio.

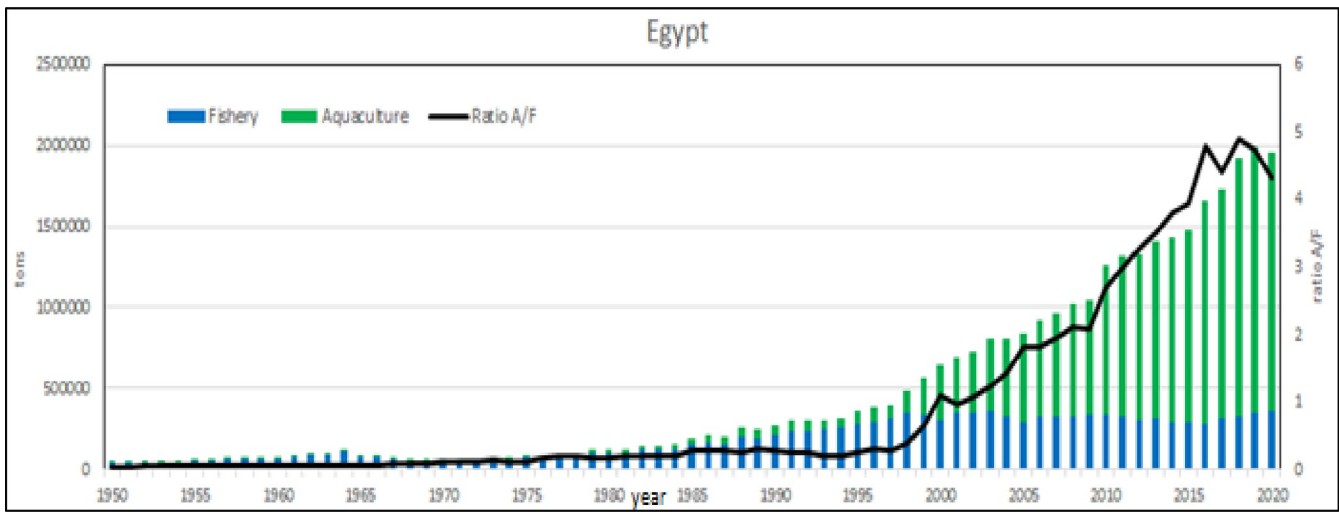

**Fig 37. Aquaculture and fishery production in Egypt from 1950 to 2020.** Black lines are the aquaculture/fishery volume ratio.

group of farming countries, green dots) are clearly separated and the increase of correlation between production, persistence (r = 0.64) and diversity (r = 0.58) indicates that countries with rooted tradition and the search for new species has promoted the development of aquaculture. The absence of correlation between country size and any aquaculture parameter confirms the idea that aquaculture productions should be analyzed in relative terms.

Previous results (Figs 38 and 39) indirectly suggest a possible influence of socio–economic factors, that have been consequently included in the last part of data analysis. Accordingly, geographic, economic and social indexes about country development, such as value of

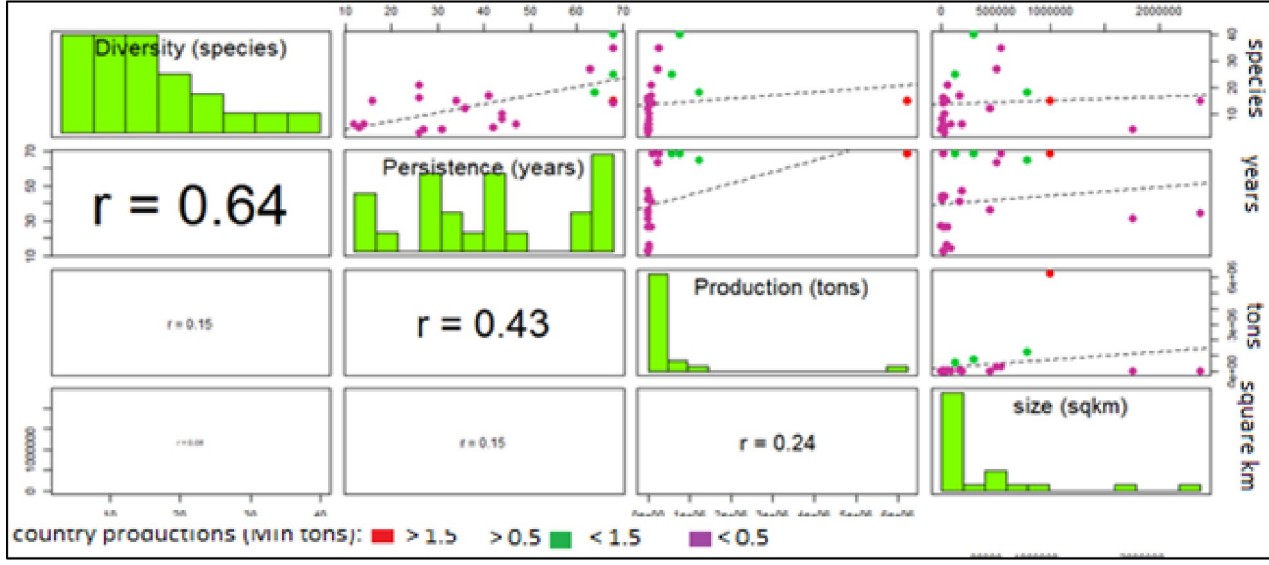

**Fig 38. Multiple correlation chart: Considering all the countries.** Countries are categorized into three main groups: high, medium and low production. Diversity (maximum number of farmed species); Persistence (maximum number of years of farming); aquaculture production (cumulate production on the last 5 years); country size (square km). Lower part of diagram: Pearson correlation coefficients (with size of character proportional to value); on the diagonal: frequency histogram of considered variable; upper part of diagram: scatterplots of countries divided into 3 categories, by production: [red dots (>1.5 Mln tons); green dots: (>0.5 and < 1.5 Mln tons); purple dots (<0.1 Mln tons) and regression line (dotted line)].

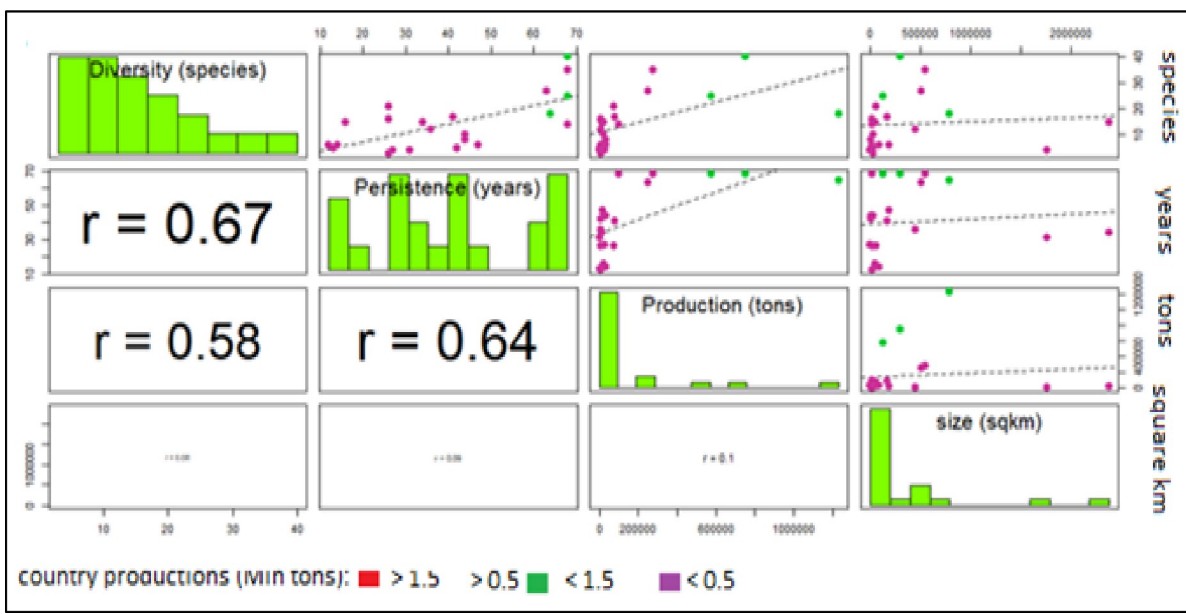

**Fig 39. Multiple correlation chart: Excluding Egypt.** Countries are categorized into three main groups: high, medium and low production. Diversity (maximum number of farmed species); Persistence (maximum number of years of farming); aquaculture production (cumulate production on the last 5 years); country size (square km). Lower part of diagram: Pearson correlation coefficients (with size of character proportional to value); on the diagonal: frequency histogram of considered variable; upper part of diagram: scatterplots of countries divided into 3 categories, by production: [red dots (>1.5 Mln tons); green dots: (>0.5 and < 1.5 Mln tons); purple dots (<0.1 Mln tons) and regression line (dotted line)].

aquaculture; Gross Domestic Product (GPD); Human Development Index (HDI); Gini coefficient (GNI); country size (km²) and coastal length have been included (Table 2), excluding Egypt. This analysis indicates that both volumes than economic value of aquaculture in the region are not clearly related with these macro-economic indexes, thus suggesting that the reasons of success in aquaculture are influenced by local conditions and that probably being aquaculture a small sector of the economies of those countries, its development cannot be foreseen by socio economic national parameters. Its effect is only visible in small countries where aquaculture is an important economic income at country level [58].

**Table 2. Between countries multiple correlation matrix (Egypt excluded).** Production (Cumulate production of aquaculture in the last 5 years, tons); Value (mean value of aquaculture in the last 5 years, Mln €)); HDI (Human Development Index); GNI (Gini coefficient); GPD (Gross domestic product); Persistence of aquaculture (years); Number of farmed species; country size (km2); Mediterranean* coastal length (km).

|  | Production Value | HDI | GNI | GPD | Persistence | Number of species | Size | Coastal length |  |
|---|---|---|---|---|---|---|---|---|---|
| Production | - | | | | | | | | |
| Value | 0,98 | - | | | | | | | |
| HDI | 0,25 | 0,27 | - | | | | | | |
| GNI | 0, 37 | 0,37 | | 0,9 | - | | | | |
| GPD | 0,52 | 0,46 | 0,42 | 0,62 | - | | | | |
| Persistence | 0,64 | 0,49 | 0,49 | 0,49 | 0,5 | - | | | |
| Number of species | 0,58 | 0,49 | 0,5 | 0,58 | 0,84 | 0,67 | - | | |
| Size | 0,01 | 0,01 | 0,25 | -0,19 | 0,01 | -0,037 | 0,01 | - | |
| Coastal length | 0,48 | 0,52 | -0,26 | 0,2 | 0,01 | 0,4 | 0,52 | -0,02 | - |

* In Spain, France, Morocco, Egypt and Turkey, only Mediterranean coast has been considered

## Discussion

### Aquaculture volumes and main species

Mediterranean aquaculture is still in an expansive phase, differently from global aquaculture production that is in a stationary phase [27]. At global level, aquaculture has overpassed capture fishery, while in the Mediterranean, aquaculture production is currently about the 50% of that of capture fishery [11, 26, 29]. The comparison between fishery and aquaculture productions shows that the future of exploitation of aquatic resources in the region will be obtained in balanced amount by aquaculture and capture fishery.

Mediterranean production growth is caused by an increase of few species (Fig 1), as Nile tilapia in Egypt [39, 59], sea bass and sea bream in Greece and middle East [2, 44, 45, 60, 66, 87, 88]. Nile tilapia is currently farmed almost completely in Egypt in brackish environment [5], while marine species are in second group together with rainbow trout. Rainbow trout has a great importance. The temporal and geographical distribution of the species in the study shows a stable group of species that represent a sort of rooted group in the region and the relevance of some minor species as eels or grass carps. Their persistent presence in the aquaculture panorama for more than 60 years is related to local or seasonal consumes. Furthermore, geographical expansion of the species shows that European sea bass and gilthead seabream are the most diffusive species in the region [36–39, 89]. Marine (including brackish water) aquaculture productions are pushed by Nile tilapia, European sea bass, gilthead seabream and mullets. European sea bass and gilthead seabream time trend are strongly correlated because of similar farming technologies and comparable market demand [82, 90, 91]. The coevolution and parallel economic growth of European sea bass and gilthead sea bream [44] suggest that these species should be considered not separate but a single aquaculture unit, a superspecies "sea bass–sea bream complex" (Figs 5 and 6) [27].

Freshwater aquaculture productions are essentially related to the farming of rainbow trout in the Northern countries, as Italy [70] France and Spain where this species is consumed in the mountain regions. Mediterranean region is traditionally considered a region based on marine species, but it should be considered that Italy, Spain and France, the inland aquaculture is a traditional activity. Fish is main aquaculture production in the region, with a limited production of marine bivalves, almost completely blue mussels.

The lack of algae aquaculture in the region, that instead at global level are main farmed species [28, 29, 74], doesn't make them necessarily candidate species. It is very well known that humans consume only a tiny fraction of the world's edible plants, for cultural reasons [92, 93].

### Aquaculture successions and species diversity

Aquaculture evolution shows a dynamic pattern that can be interpreted as a succession of aquaculture communities of species that progressively have evolved starting from pioneering species toward the species farmed in the modern period, which constitute an anthropogenic climax (Fig 18). Ecological successions have been deeply investigated in ecology [94] and the application here of this model clearly suggests the idea of colonization. Therefore, aquaculture acquires the form of an anthropogenic colonization of natural resources. The similitude between the ecological succession of colonizing species in a forest ecosystem and the succession of species in the Mediterranean is impressive (Fig 18). It cannot be considered a mere coincidence but rather an evolutionary convergence. Selected farmed species have progressively colonized the Mediterranean region. Accordingly, pioneer, intermediate and climax species can be observed. This ecological approach makes clearer the evolution of aquaculture and

the current composition of farmed species, but more interestingly, could give an original interpretative key for its future.

This interpretation is certainly a simplification as it synthetizes the evolution of 21 species along a period of 70 years, but it has the great advantage to give a quantitative and univocal description. Mediterranean aquaculture evolution is cyclical, and it has been characterized by a progressive shift from freshwater to marine aquaculture, with a series of 3 successive cycles of about 20 years of duration each. Considering the huge production of Nile tilapia together with gilthead sea bream and European sea bass in the last period, (Figs 16–18) it cannot be stated that in the Mediterranean region there is a situation of increase of mean trophic level of farmed species [6, 54, 82].

The utilize of alternative measures of species diversity in this research, has made evident that species richness is meaningless for the future development of aquaculture in the region, thus confirming global data [27]. Tree to five species have produced the 80% of annual production for almost all the investigated period. This means that the "skeleton" of Mediterranean aquaculture is substantially stationary, but increase of diversity is a superficial aspect of aquaculture that essentially regards minor species.

## Conclusions: Mediterranean, a sea of contradictions

In these last 70 years, Mediterranean region is passing through a transition phase in the exploitation of aquatic resources: from an exploitation exclusively based on utilize of natural resources (capture fishery) to a future balanced utilize between fishery and aquaculture that will reach fishery volume in next 20 years. In evolutionary terms, Mediterranean region is encompassing a transition from direct to deferred exploitation of resources. Even Mediterranean aquaculture corresponds to the 2,2% of world aquaculture production, at regional level, its production currently is equivalent to the 50% of total Mediterranean fishery production.

Mediterranean aquaculture has been characterized by an exorbitant growth of few species and few countries. It will continue to grow in the next future together with fish consumptions. Egypt is the dominant country and Nile tilapia is the dominant species.

Nile tilapia and Egypt will dominate the future of Mediterranean aquaculture, in the short and middle term. Egypt before 2030 is expected to produce more than 2 Mln tons of aquaculture Therefore, Mediterranean aquaculture is dominated by a fish species farmed in one single country, Egypt. In several countries, aquaculture has huge potentiality of development and it could growth with a factor of 5 or more, based on the ratio with capture fishery on country size. However, Malta that is the most productive country in relative terms, cannot be considered a suitable model of aquaculture development. The largest countries both in relative than in absolute terms (Malta and Egypt respectively) cannot be considered as forecasting models for the entire region.

Even if Nile tilapia dominates for production volumes, the most common species in the region are European sea bass and gilthead sea bream that are real trans-Mediterranean species and promising for a future expansion.

The evolution of aquaculture in a certain region is characterized by the succession of successful group of species, a succession that evolves toward an anthropogenic climax stage. Mediterranean species climax community is composed almost completely by Nile tilapia, European sea bass and gilthead seabream that are main component of that community. The idea previously proposed of a farming up in the food web [6, 54] should be tempered by the consideration of Nile tilapia production.

The future of aquaculture in the Mediterranean region will rely on 4 or 5 species that will produce the 80% of annual production. However, the total number of farmed species will

continue to grow in a quasi-exponential way (Fig 22). Minor species constitute a kind of superficial layer that conceals the real essence of aquaculture diversity. The continuous progress of aquaculture techniques and the rapidity of diffusion of innovation make aquaculture a cyclical activity related with technology, however in the Mediterranean region is not the only interpretative key, as Egypt aquaculture is largely based on extensive techniques [95].

In evolutionary biology terms, the diffusion on technological innovation is the manifestation of cultural evolution and its overlapping with biological evolution. Socio-political and legal aspects are crucial for the development of aquaculture [23, 74, 95], and here have been neglected for the utilization of an evolutionary approach and for sake of conciseness, therefore this analysis is necessarily uncompleted. However, the importance of political factors has indirectly emerged in several point of this paper.

This paper has also hopefully shown that the inclusion of a bit of theoretical ecology and evolutionary biology, as species communities, geographical expansion compared with a colonization of new environment, can hopefully benefit the discussion on aquaculture and its perspectives in this region and wherever.

## Supporting information

**S1 Video. Mediterranean aquaculture video clip.**
(GIF)

## Author Contributions

**Conceptualization:** Benedetto Sicuro.

**Data curation:** Benedetto Sicuro.

**Formal analysis:** Benedetto Sicuro.

**Writing – original draft:** Benedetto Sicuro.

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
