## [Decision Letter · Decision Letter 0]

17 Feb 2023

PONE-D-22-29781The evolution of aquaculture in the Mediterranean region: an anthropogenic climax stage?PLOS ONE

Dear Dr. Sicuro,

Thank you for submitting your manuscript to PLOS ONE. After careful consideration, we feel that it has merit but does not fully meet PLOS ONE’s publication criteria as it currently stands. Therefore, we invite you to submit a revised version of the manuscript that addresses the points raised during the review process.

Although the reviewers have underlined the interest of the study, they have also highlighted some critical issues regarding the methods (e.g. they should be better specified) and the analysis (e.g. about the production value, a more in-depth analysis of the Egyptian production, about the shellfish production) . Below you will find the specific comments to which I ask you to respond to all exhaustively.

We look forward to receiving your revised manuscript.

Kind regards,

Pierluigi Carbonara, PhD

Academic Editor

PLOS ONE

Journal Requirements:

3. We note that Figures 2.1, 2.2 and 2.5 in your submission contain map images which may be copyrighted. All PLOS content is published under the Creative Commons Attribution License (CC BY 4.0), which means that the manuscript, images, and Supporting Information files will be freely available online, and any third party is permitted to access, download, copy, distribute, and use these materials in any way, even commercially, with proper attribution. For these reasons, we cannot publish previously copyrighted maps or satellite images created using proprietary data, such as Google software (Google Maps, Street View, and Earth). For more information, see our copyright guidelines: http://journals.plos.org/plosone/s/licenses-and-copyright.

a. You may seek permission from the original copyright holder of Figures 2.1, 2.2 and 2.5 to publish the content specifically under the CC BY 4.0 license.  

Reviewers' comments:

Reviewer's Responses to Questions

**Comments to the Author**

1. Is the manuscript technically sound, and do the data support the conclusions?

Reviewer #1: Yes

Reviewer #2: Partly

2. Has the statistical analysis been performed appropriately and rigorously? 

Reviewer #1: Yes

Reviewer #2: No

3. Have the authors made all data underlying the findings in their manuscript fully available?

Reviewer #1: Yes

Reviewer #2: No

4. Is the manuscript presented in an intelligible fashion and written in standard English?

Reviewer #1: Yes

Reviewer #2: Yes

5. Review Comments to the Author

Reviewer #1: Great amount of effort on collecting and analyzing all these data for the Mediterranean aquaculture. I would suggest some minor corrections/ amendments:

line 43 please change the order of the word "here" with the word "analyzed"

line 58 please delete the dot after 2017

Reference 71 is missing so update the reference numbers

In figures 1.1, 1.2, 1.5.1, 12.2 please include at the caption what is the dot line in the figure

In figure 1.2 please explain in the caption what is the black line

In figures 2.3 and 2.4 please explain in the caption what is the black and the blue lines

In figure 2.7 please change the font size of the legends in the figures because it is rather difficult to read them as it is now. Additionally, describe in the caption what represents the black line in the figure.

Finally, in the methodology, I would suggest to give a bit more information about the selected statistical tools of the data analysis.

Reviewer #2: Review about manuscript. “The evolution of aquaculture in the Mediterranean region: an anthropogenic climax stage?”

The author presents a comprehensive analysis of production data and statistics on aquaculture in the Mediterranean. This provides new insights on what could be the evolution of the sector.

There are, however, some very important points to improve in the analysis presented here. Most of them are related to methodological aspects. Among them:

i) The author should better clarify and specify in Materials and Methods what is meant by the term Mediterranean region in this work. Which statistics and countries are considered under this term. It is clear that continental aquaculture (trout) is considered, but not if the statistics include aquaculture in the Atlantic (oysters, mussels, turbot, etc.) or in the Black Sea.

ii) It is striking that the analysis of FAO statistics stops at the year 2017, when from March 2022 there is the dataset with 3 more years statistics (up to 2020) is availabe. This update is essential, as there have been significant variations in recent years, which could have a major impact on the trends.

iii) Another aspect is that it is highly advisable to carry out an analysis of production in the Mediterranean considering the value of production. This would help the reader to gain knowledge on the reasons being the development of the sector. For example, while in Egypt, the leading country in the Mediterranean, the main driving force for development is the need to provide animal protein of high nutritional value and affordable prices (tilapia, mullets), in the rest of the Mediterranean that is different, and thus higher value species are produced.

iv) The author analyses aquaculture in Egypt separately, which is appropriate as it is the main producer country in the region. However, it would be advisable to provide a more in-depth description of aquaculture development in this country, since a very significant amount of production (mullets) is carried out in brackish waters, and this is reflected in the statistics. In fact Egyptian mullets production may account for near 20% production in the regions.

v) Please, revise the sentence in lines 168 and 169. “Fish farming is the main production in the Mediterranean [53, 6, 54], in fact in 2017 accounted for the 96% of entire aquaculture production in the region.” This must be a mistake, as bivalve production may be around 20% of the whole production.

vi) Besides the need of updating the statistics, considering the latest dataset from March 2022 (FAO-FishStatJ. Aquaculture Production (Quantities and values) 1950-2020 (Release date: March 2022), the author may profit of the occasion to include some recent relevant publications. i.e.:

Llorente, I., Fernandez-Polanco, J., Baraibar-Diez, E., Odriozola, M. D., Bjorndal, T., Asche, F., … Basurco, B. (2020). Assessment of the economic performance of the seabream and seabass aquaculture industry in the European Union. Marine Policy, 117, 103876. https://doi.org/ 10.1016/j.marpol.2020.103876

COMMUNICATION FROM THE COMMISSION TO THE EUROPEAN PARLIAMENT, THE COUNCIL, THE EUROPEAN ECONOMIC AND SOCIAL COMMITTEE AND THE COMMITTEE OF THE REGIONS Strategic guidelines for a more sustainable and competitive EU aquaculture for the period 2021 to 2030

Fernandez Sanchez, José Luís & Llorente, Ignacio & Fernandez Polanco, Jose. (2023). Profitability differences in aquaculture firms of the Nordic and Mediterranean-EU regions. Aquaculture Economics & Management. 1-17. 10.1080/13657305.2022.2163721.

Best regards

6. PLOS authors have the option to publish the peer review history of their article (what does this mean?). If published, this will include your full peer review and any attached files.

Reviewer #1: No

Reviewer #2: No

---

## [Author Response · Author response to Decision Letter 0]

23 Jun 2023

5. Review Comments to the Author

Reviewer #1: Great amount of effort on collecting and analyzing all these data for the Mediterranean aquaculture. 

Thank you very much for your comments.

I would suggest some minor corrections/ amendments:

line 43 please change the order of the word "here" with the word "analyzed"

productions [23], but the 67 years of production data analyzed here represent the entire history of

line 58 please delete the dot after 2017

OK

Reference 71 is missing so update the reference numbers

I added citation 71 in the text 

In figures 1.1, 1.2, 1.5.1, 12.2 please include at the caption what is the dot line in the figure

I corrected figure 1.1 legend as suggested:

Dotted line: polynomial regression line, with equation and regression coefficients.

I uploaded most recent aquaculture data until 2020. I added economic value of aquaculture from 1984 to 2020 (FAO data on economic value are only recorded from 1984)

Fig. 1.1. a) Mediterranean aquaculture productions and annual growth rate from 1950 to 2020. b) Mediterranean aquaculture economic value and annual growth rate from 1984 to 2020. Dotted line: polynomial regression line, with equation and regression coefficients. (FAO. 2022. Fishery and Aquaculture Statistics. Global aquaculture production 1950-2020 (FishStatJ). In: FAO Fisheries and Aquaculture Division [online]. Rome. Updated 2022. ww.fao.org/fishery/statistics/software/fishstatj/en)

In figure 1.2: dotted line, legend correct as follows:

Fig 1.2 Fishery and aquaculture productions in the Mediterranean. Dotted line on the top: polynomial and linear regression lines, with relative equations and regression coefficients.

In figure 1.5.1: dotted line, legend correct as follows:

Dotted lines: polynomial regression lines, with relative equations and regression coefficients.

In figure 1.2 please explain in the caption what is the black line

Fig 1.2 Fishery and aquaculture productions in the Mediterranean. Black line, bottom: ratio aquaculture/fishery volumes. Dotted line, top: polynomial and linear regression lines, with relative equations and regression coefficients.

In figures 2.3 and 2.4 please explain in the caption what is the black and the blue lines

Fig 2.3 – Aquaculture value (blue line) and GDP pro capita (black line) (considering the 3 previous zones in aquaculture zonation obtained in fig 2.1). In the corner, on the left, the Pearson correlation coefficient (r).

Fig 2.4 – Aquaculture production (blue line) and sea food* consumption pro capita (kg/year) (black line) (considering the 3 previous zones in aquaculture zonation obtained in fig 2.1). In the corner, on the left, the Pearson correlation coefficient (r). (*sea food includes: cephalopods, crustaceans, demersal fish, fish body oil, fish liver oil, freshwater fish, marine fish, other mollusks, other pelagic fish, according all the items contained in the database: https://www.fao.org/faostat/en/#data/CL)

In figure 2.7 please change the font size of the legends in the figures because it is rather difficult to read them as it is now. Additionally, describe in the caption what represents the black line in the figure.

Fig 2.7 Aquaculture and fishery production in Malta, Egypt and Israel from 1950 to 2020. Black lines are the aquaculture/fishery volume ratio

I increased font size, I included description of black line. I also downloaded data until 2020 and updated the figures 

Finally, in the methodology, I would suggest to give a bit more information about the selected statistical tools of the data analysis.

I inserted some short sentences in the section, to clarify the statistical methods, as follows:

Both multivariate and regression methods have been used for data elaboration.

Dendrograms have been calculated only on the rows that are fish species, as the focus of the data analysis was the study of temporal succession of farmed species in the Mediterranean. Multiple correlation plot has been obtained with R package "Performance Analytics", ". In the multiple correlation plots, frequency histograms have been added.

 

Reviewer #2: Review about manuscript. “The evolution of aquaculture in the Mediterranean region: an anthropogenic climax stage?”

The author presents a comprehensive analysis of production data and statistics on aquaculture in the Mediterranean. This provides new insights on what could be the evolution of the sector.

There are, however, some very important points to improve in the analysis presented here. Most of them are related to methodological aspects. Among them:

Dear Reviever #2,

Thank you for your comments, I will try to reply them as best as I can.

i) The author should better clarify and specify in Materials and Methods what is meant by the term Mediterranean region in this work. Which statistics and countries are considered under this term.

In this study I utilized all the countries of Mediterranean Sea represented in FAOSTAT databases: Malta, Egypt, Israel, Greece, Cyprus, Italy, Turkey, Croatia, Albania, France, Spain, Tunisia, Lebanon, Bosnia, Slovenia, Montenegro, Syria, Morocco, Algeria, Libya, and Palestine (indicated in tab. 2.1), thus respecting most recent geopolitical situation (in 2017). 

I have excluded non-Mediterranean regions of countries which coastlines are not exclusively Mediterranean (i.e. Spain, France, Morocco, Egypt and Turkey) as indicated in legend of tab 2.2 

I have also considered socio-political changes occurred in the course of 68 years analyzed, in particular for countries corresponding to ex-Yugoslavia (as indicated in figure 2.2)

It is clear that continental aquaculture (trout) is considered, but not if the statistics include aquaculture in the Atlantic (oysters, mussels, turbot, etc.) 

I have considered all the species farmed in all the considered countries during the 68 years contained in FAO databases

or in the Black Sea.

I did not consider Black Sea, I focused my analysis on Mediterranean Sea

 

ii) It is striking that the analysis of FAO statistics stops at the year 2017, when from March 2022 there is the dataset with 3 more years statistics (up to 2020) is availabe. This update is essential, as there have been significant variations in recent years, which could have a major impact on the trends.

I uploaded the most recent FAO databases of aquaculture productions and capture fishery (FAO. 2022. Fishery and Aquaculture Statistics. Global capture production 1950-2020 (FishStatJ). In: FAO Fisheries and Aquaculture Division [online]. Rome. Updated 2022. www.fao.org/fishery/statistics/software/fishstatj/en) and I consequently uploaded figures with the most recent available data.

During the second elaboration of data from 1950 to 2020, I also found 2 mistakes I did in the previous data elaboration in the figures 1.3 and 2.6. I corrected them and I included a correct interpretation. 

Fig. 1.1a Mediterranean aquaculture productions and annual growth rate from 1950 to 2020. (FAO. 2022. Fishery and Aquaculture Statistics. Global aquaculture production 1950-2020 (FishStatJ). In: FAO Fisheries and Aquaculture Division [online]. Rome. Updated 2022. www.fao.org/fishery/statistics/software/fishstatj/en)

I uploaded the most recent FAO databases of aquaculture economic value, that starts from 1984

Fig 1.1b Mediterranean aquaculture economic value and annual growth rate from 1984 to 2020 (FAO. 2022. Fishery and Aquaculture Statistics. Global aquaculture production 1950-2020 (FishStatJ). In: FAO Fisheries and Aquaculture Division [online]. Rome. Updated 2022. www.fao.org/fishery/statistics/software/fishstatj/en)

Fig 1.2 Fishery and aquaculture productions in the Mediterranean

Figure 1.2.2 Evolution of two main aquaculture sectors: marine and freshwater aquaculture. Trend and annual growth rate

Fig 1.2.3 Fish and bivalve farming in the Mediterranean region. Trend and growth rate

Fig. 1.2.4 Aquaculture productions of Nile tilapia, mullets, European sea bass and gilthead seabream.

Fig 1.3 Capture fishery and aquaculture annual productions (considering only Mediterranean coastline of respective countries)

I apologize. I did a mistake in the previous data as I did not exclude fishery data from the extra Mediterranean coasts. Part of the Morocco, Spain France, Turkey and Egypt coasts are in Atlantic and Pacific oceans. Consequently, the fishery production resulted higher than production solely relative to Mediterranean Sea. However, the interpretation of data is comparable with the previous one. 

Fig 1.4.1 – Heatmaps of main aquaculture species in the Mediterranean (A: original data; B: scaled data). Data scaling: x scaled = (x-μ)/s.d. (μ=mean; s.d.=standard deviation)

Fig. 1.4.2 Comparison between succession of farmed species in the Mediterranean in 3 stages (A and C) and a forest succession in 6 stages (B) (from Wikipedia https://commons.wikimedia.org/w/index.php?curid=13274746, modified)

Fig. 1.5.1 Aquaculture diversity in the Mediterranean region, considering its main components: main, relevant and minor species. A: Pareto Index trend (ratio main/relevant species); B: evolution of different components

Fig 1.5.2 Aquaculture diversity in the Mediterranean region. Shannon Index

Fig. 2.2 Map aquaculture persistence (years of farming in each country) *countries corresponding to ex-Yugoslavia have been counted considering the historic change of national boundaries

Fig. 2.6 Geographical expansion of aquaculture in the Mediterranean region. In red the number of countries producing the 80% of annual production. Pareto index (%) = (number of countries producing the 80% of annual production/total countries aquaculture producing)

I apologize, in this figure I fond an error I have done in the previous data elaboration. Pareto index is 23% for the entire considered period, even if in the first years there are some oscillation caused by low number of countries producing the 80% of total production. 

This value is closer to that relative to species diversity (Fig. 1.5.1) and also with a previous study on world aquaculture diversity (Sicuro B. World aquaculture diversity: origins and perspectives. Rev. Aquac. 2021; 13, 1619-1634 https://doi.org/10.1111/raq.12537). 

Fig 2.7 Aquaculture and fishery production in Malta, Egypt and Israel from 1950 to 2020

 

iii) Another aspect is that it is highly advisable to carry out an analysis of production in the Mediterranean considering the value of production. This would help the reader to gain knowledge on the reasons being the development of the sector. For example, while in Egypt, the leading country in the Mediterranean, the main driving force for development is the need to provide animal protein of high nutritional value and affordable prices (tilapia, mullets), in the rest of the Mediterranean that is different, and thus higher value species are produced.

I added the evolution of economic value of Mediterranean aquaculture from 1984 to 2020, including annual growth rate and polynomial model of regression 

Fig 1.1b Total aquaculture from 1984 to 2020 (trend and growth rate) (FAO. 2022. Fishery and Aquaculture Statistics. Global aquaculture production 1950-2020 (FishStatJ). In: FAO Fisheries and Aquaculture Division [online]. Rome. Updated 2022. www.fao.org/fishery/statistics/software/fishstatj/en)

 

iv) The author analyses aquaculture in Egypt separately, which is appropriate as it is the main producer country in the region. However, it would be advisable to provide a more in-depth description of aquaculture development in this country, since a very significant amount of production (mullets) is carried out in brackish waters, and this is reflected in the statistics. In fact Egyptian mullets production may account for near 20% production in the regions.

I added following figure (Fig. 2.8), considering main species of Egyptian aquaculture 

Figure 2.7b Aquaculture productions in Egypt (main species) from 1950 to 2020

 

v) Please, revise the sentence in lines 168 and 169. “Fish farming is the main production in the Mediterranean [53, 6, 54], in fact in 2017 accounted for the 96% of entire aquaculture production in the region.” This must be a mistake, as bivalve production may be around 20% of the whole production.

I checked data, I re-made elaborations, I introduced a figure on Egyptian aquaculture (Fig 2.7b) and I consequently correct as follows:

Fish farming is the main production in the Mediterranean [53, 6, 54], in fact in 2020 accounted for more than the 90% of entire aquaculture production in the region. Within fish species, Mediterranean aquaculture has 4 species with potential further growth: Nile tilapia, European sea bass, gilthead seabream and in less extent, mullets (Fig. 1.2.4)

Following: carps and mullets productions in Egypt (not introduced in the manuscript)

 

vi) Besides the need of updating the statistics, considering the latest dataset from March 2022 (FAO-FishStatJ. Aquaculture Production (Quantities and values) 1950-2020 (Release date: March 2022), the author may profit of the occasion to include some recent relevant publications. i.e.:

Llorente, I., Fernandez-Polanco, J., Baraibar-Diez, E., Odriozola, M. D., Bjorndal, T., Asche, F., … Basurco, B. (2020). Assessment of the economic performance of the seabream and seabass aquaculture industry in the European Union. Marine Policy, 117, 103876.https://doi.org/10.1016/j.marpol.2020.103876

Added as 88th reference

COMMUNICATION FROM THE COMMISSION TO THE EUROPEAN PARLIAMENT, THE COUNCIL, THE EUROPEAN ECONOMIC AND SOCIAL COMMITTEE AND THE COMMITTEE OF THE REGIONS Strategic guidelines for a more sustainable and competitive EU aquaculture for the period 2021 to 2030

Brussels, 12.5.2021

Added as 82th reference

Fernandez Sanchez, José Luís & Llorente, Ignacio & Fernandez Polanco, Jose. (2023). Profitability differences in aquaculture firms of the Nordic and Mediterranean-EU regions. Aquaculture Economics & Management. 1-17. 10.1080/13657305.2022.2163721.

Best regards

6. PLOS authors have the option to publish the peer review history of their article (what does this mean?). If published, this will include your full peer review and any attached files.

Do you want your identity to be public for this peer review? For information about this choice, including consent withdrawal, please see our Privacy Policy.

Reviewer #1: No

Reviewer #2: No

---

## [Decision Letter · Decision Letter 1]

18 Aug 2023

The evolution of aquaculture in the Mediterranean region: an anthropogenic climax stage?

PONE-D-22-29781R1

Dear Dr. Sicuro,

We’re pleased to inform you that your manuscript has been judged scientifically suitable for publication and will be formally accepted for publication once it meets all outstanding technical requirements.

Kind regards,

Pierluigi Carbonara, PhD

Academic Editor

PLOS ONE

Additional Editor Comments (optional):

Reviewers' comments:

Reviewer's Responses to Questions

**Comments to the Author**

1. If the authors have adequately addressed your comments raised in a previous round of review and you feel that this manuscript is now acceptable for publication, you may indicate that here to bypass the “Comments to the Author” section, enter your conflict of interest statement in the “Confidential to Editor” section, and submit your "Accept" recommendation.

Reviewer #1: All comments have been addressed

2. Is the manuscript technically sound, and do the data support the conclusions?

Reviewer #1: Yes

3. Has the statistical analysis been performed appropriately and rigorously? 

Reviewer #1: Yes

4. Have the authors made all data underlying the findings in their manuscript fully available?

Reviewer #1: Yes

5. Is the manuscript presented in an intelligible fashion and written in standard English?

Reviewer #1: Yes

6. Review Comments to the Author

Reviewer #1: (No Response)

7. PLOS authors have the option to publish the peer review history of their article (what does this mean?). If published, this will include your full peer review and any attached files.

Reviewer #1: No

---

## [Editor Report · Acceptance letter]

7 Sep 2023

PONE-D-22-29781R1 

The evolution of aquaculture in the Mediterranean region: an anthropogenic climax stage? 

Dear Dr. Sicuro:

I'm pleased to inform you that your manuscript has been deemed suitable for publication in PLOS ONE. Congratulations! Your manuscript is now with our production department. 

Kind regards, 

on behalf of

Dr. Pierluigi Carbonara 

Academic Editor

PLOS ONE